# Equine Musculoskeletal Pathologies: Clinical Approaches and Therapeutical Perspectives—A Review

**DOI:** 10.3390/vetsci11050190

**Published:** 2024-04-26

**Authors:** Inês L. Reis, Bruna Lopes, Patrícia Sousa, Ana C. Sousa, Ana R. Caseiro, Carla M. Mendonça, Jorge M. Santos, Luís M. Atayde, Rui D. Alvites, Ana C. Maurício

**Affiliations:** 1Departamento de Clínicas Veterinárias, Instituto de Ciências Biomédicas de Abel Salazar (ICBAS), Universidade do Porto (UP), Rua de Jorge Viterbo Ferreira, n° 228, 4050-313 Porto, Portugal; up199900642@edu.icbas.up.pt (I.L.R.); up201304418@edu.icbas.up.pt (B.L.); up201403005@edu.icbas.up.pt (P.S.); up201502681@edu.fep.up.pt (A.C.S.); cmmendonca@icbas.up.pt (C.M.M.); jmpsantos@icbas.up.pt (J.M.S.); lmathayde@icbas.up.pt (L.M.A.); malvites@icbas.up.pt (R.D.A.); 2Centro de Estudos de Ciência Animal (CECA), Instituto de Ciências, Tecnologias e Agroambiente da Universidade do Porto (ICETA), Rua D. Manuel II, Apartado 55142, 4051-401 Porto, Portugal; rita.caseiro@euvg.pt; 3Associate Laboratory for Animal and Veterinary Science (AL4AnimalS), 1300-477 Lisboa, Portugal; 4Departamento de Ciências Veterinárias, Cooperativa de Ensino Superior Politécnico e Universitário (CESPU), Instituto Universitário de Ciências da Saúde (IUCS), Avenida Central de Gandra 1317, 4585-116 Gandra, Portugal; 5Veterinary Sciences Department, University School Vasco da Gama (EUVG), Avenida José R. Sousa Fernandes, Lordemão, 3020-210 Coimbra, Portugal; 6Vasco da Gama Research Center (CIVG), University School Vasco da Gama (EUVG), Avenida José R. Sousa Fernandes, Lordemão, 3020-210 Coimbra, Portugal; 7Campus Agrário de Vairão, Centro Clínico de Equinos de Vairão (CCEV), Rua da Braziela n° 100, 4485-144 Vairão, Portugal

**Keywords:** conservative therapies, equine, hemoderivatives, musculoskeletal, osteoarthritis, physiotherapy, regenerative therapies, sports medicine, stem cell, tendinitis

## Abstract

**Simple Summary:**

In the current era, sport horses face the challenge of increasingly rigorous workouts, overtaking, and, at times, certain physiological thresholds, leaving them susceptible to injury. This shift underscores the significance of prioritizing the exercise preparation quality and post-care to proactively mitigate the risk of injuries. Despite these measures, injuries may still occur, thus requiring a nuanced understanding of the most effective therapeutic approaches for various types of lesions. In the dynamic field of equine sports medicine, keeping abreast of the expansive therapeutic options proves challenging, especially when aiming to address injuries comprehensively, restore organ function, and sustain the horse’s athletic life. The present endeavor seeks to elucidate the array of available therapies, encompassing both conservative and regenerative methods, for the effective management of musculoskeletal injuries in sport horses.

**Abstract:**

Musculoskeletal injuries such as equine osteoarthritis, osteoarticular defects, tendonitis/desmitis, and muscular disorders are prevalent among sport horses, with a fair prognosis for returning to exercise or previous performance levels. The field of equine medicine has witnessed rapid and fruitful development, resulting in a diverse range of therapeutic options for musculoskeletal problems. Staying abreast of these advancements can be challenging, prompting the need for a comprehensive review of commonly used and recent treatments. The aim is to compile current therapeutic options for managing these injuries, spanning from simple to complex physiotherapy techniques, conservative treatments including steroidal and non-steroidal anti-inflammatory drugs, hyaluronic acid, polysulfated glycosaminoglycans, pentosan polysulfate, and polyacrylamides, to promising regenerative therapies such as hemoderivatives and stem cell-based therapies. Each therapeutic modality is scrutinized for its benefits, limitations, and potential synergistic actions to facilitate their most effective application for the intended healing/regeneration of the injured tissue/organ and subsequent patient recovery. While stem cell-based therapies have emerged as particularly promising for equine musculoskeletal injuries, a multidisciplinary approach is underscored throughout the discussion, emphasizing the importance of considering various therapeutic modalities in tandem.

## 1. Introduction

In equine sports medicine, musculoskeletal lesions such as tendonitis, desmitis, osteoarticular lesions/defects, and muscular strains stand as significant challenges. These conditions often result in a partial or complete loss of performance, jeopardizing the athlete’s sporting career. Beyond the immediate impact on performance, these injuries can have far-reaching consequences, affecting the overall well-being of the horse [1,2]. The repercussions extend to individual health, sporting activity, and carry social and economic implications, making the management and prevention of such musculoskeletal issues crucial in the field of equine sports [3,4,5].

Each of these injuries presents unique characteristics and are a challenge in terms of diagnosis, treatment, and rehabilitation. Understanding and addressing these issues is vital for maintaining the health and longevity of equine athletes.

The present work provides an overview of prevalent musculoskeletal pathologies, explores complementary diagnostic exams, and reviews the existing therapeutic options for managing pain, inflammation, and the healing or regeneration of affected musculoskeletal structures. The current prospect offers a diverse range of therapeutic options, each varying in ease of use, availability, economic considerations, mode of action, effectiveness, and clinical outcomes.

Initial treatment protocols emphasize conservative interventions aimed at alleviating clinical signs, slowing progression, and potentially facilitating tissue repair through fibrosis or scar formation [6]. These interventions span both non-pharmacological and pharmacological approaches, encompassing a spectrum of exercises and physical modalities (e.g., laser therapy, ultrasound, electrotherapy, and shock waves) as well as pharmacological options including anti-inflammatories, viscosupplementation, and bisphosphonates. Surgical techniques such as tendon splitting are also employed for conditions such as tendonitis to provide relief from the clinical signs and induce tissue repair.

More recently, the focus has shifted toward achieving tissue regeneration rather than mere repair. Several pro-regenerative therapeutic options have emerged, and others are currently in development. Termed orthobiologics, a subdivision of regenerative medicine, with a specific emphasis on addressing musculoskeletal conditions, these therapies are based on biological substances to promote regeneration in the tendons, muscles, and joints [7]. Two major categories within orthobiologics are hemoderivatives and stem cell-based therapies. The composition, distinctions, beneficial contributions, advantages offered by each, and the appropriate scenarios for their application will be elucidated through the article. The overarching objective of these regenerative therapies is to preserve organ function and facilitate the restoration of native physiological capacity in musculoskeletal diseases.

## 2. Clinical Examination

A complete history and a comprehensive clinical and orthopedic examination hold paramount importance in the field of orthopedic medicine to accurately diagnose musculoskeletal conditions [8]. The findings from a clinical and orthopedic examination guide the formulation of an appropriate plan of complementary diagnostic exams and a tailored treatment strategy. Additionally, the examination aids in identifying any underlying issue contributing to musculoskeletal problems. Orthopedic examination must include inspection, palpation, percussion, hoof testers, and dynamic analysis of the horse. This holistic approach ensures that all relevant factors are considered for comprehensive care [9]. Regular clinical examinations can help identify risk factors or early signs of musculoskeletal issues. This allows for the implementation of preventive measures to mitigate the progression of conditions or prevent future injuries. Follow-up examinations are essential to monitor the progress of treatment and adjusting the interventions as needed. This constant process ensures that the patient’s musculoskeletal health is continually assessed and managed effectively. In summary, equine clinical examinations are essential for maintaining the health, well-being, and performance of horses [10].

## 3. Complementary Diagnostic Exams

To accurately diagnose musculoskeletal pathologies, comprehensive physical and orthopedic examinations, along with detailed complementary diagnostic (CD) tests, are imperative. CD tools encompass a broad-spectrum including flexion tests, perineural nerve blocks, intra-articular anesthesia, and various imaging techniques. The selection of these tools depends on factors such as accuracy, accessibility, cost, and the ability of individuals or professionals to interpret the images. For the diagnosis of musculoskeletal injuries, available imaging techniques comprise radiographs (X-ray), ultrasound (U/S) images, arthroscopy, magnetic resonance imaging (MRI), computed tomography (CT), scintigraphy and positron emission tomography (PET) scans. Each of these methods offers unique advantages and considerations in the diagnostic process.

Perineural nerve blocks and intra-articular anesthesia aid in the determination of the pain localization area [11]. 

Radiography stands as a non-invasive and primary diagnostic tool, offering crucial insights into several aspects. It enables the identification of significant information, encompassing the diagnosis of evident bone fractures, defects in articular cartilage, and indications of OA. In radiological OA diagnosis, three primary findings are prominent: the presence of osteophytes, increased subchondral density, and the eventual narrowing of the joint space in its advanced stages [12].

Nevertheless, during the early stages of a lesion, the observation of radiological evidence becomes challenging, potentially leading to inappropriate exclusion or insufficient information regarding the actual extent of the lesions. Moreover, when assessing complex joints, radiography faces limitations as it provides a two-dimensional (2D) representation of three-dimensional (3D) structures. This limitation arises because multiple soft tissues and bony structures are superimposed, making it difficult to evaluate them individually. Consequently, radiography may not be the most favorable method for the assessment of soft tissues [13,14]. Nonetheless, when dealing with cases involving joint or limb distension, radiography is recommended as an initial diagnostic procedure. In instances of acute tendinopathy, desmopathy, or enthesopathy without fragmentation of the insertional surface, radiography may primarily indicate the presence of soft tissue swelling [15]. 

Ultrasonography has emerged as the preferred technique for diagnosing, assessing, and documenting tendon and ligament lesions, offering valuable insights into therapeutic and healing progresses. The ultrasonographic method for evaluating the pastern region, specifically for palmar/plantar tendon/ligament assessment, is roughly documented in the existing literature [16,17,18]. A thorough ultrasound assessment of soft tissue injuries is very important to accurately diagnose these lesions [7]. However, in conditions of intra-articular soft tissue lesions, it does not allow for the visualization of structures located deep to the bone [14,19,20]. Since 1990, ultrasonography has been used to complement radiography in cartilage defects diagnosis [21]. Ultrasound has become an essential diagnostic tool as it provides precise information on the synovial membrane and fluid, articular cartilage, subchondral bone, joint margins, ligaments and menisci as well as on the periarticular structures [21]. It requires good ultrasound devices and a strict standardization of the approach technique to every joint. A substantial anatomical knowledge of the equine locomotor system is crucial to warrant the sensitivity and specificity of the diagnostic ultrasonography of joint structures [21,22]. 

In cases of joint lesions, arthroscopy has proven to be a valuable diagnostic tool with dual functionality for both diagnosis and treatment. This invasive technique enables the intra-articular visualization of the cartilage surface, synovia, and ligaments, offering information on the characteristics of cartilage defects and allowing for grading. Arthroscopy is particularly adept at assessing early, subtle cartilage lesions. However, it should be noted that it has limitations, as it may underestimate the extent of certain lesions and overestimate others. This is attributed to its inability to provide a representative image of the entire articular surface [23,24]. At the same time, it also grants the treatment of some lesions such as the removal of cartilage fragments and joint/synovial debris, osteochondral *curettage* as well as sample collection.

Currently, MRI is the optimal method for the evaluation and monitoring of tendon/ligament and articular cartilage health, providing cross sectional images. While it is a non-invasive assessment technique, it can detect soft tissue changes and assess the cartilage morphology. It also provides volumetric and whole joint assessment without ionizing radiation. This technique is able to non-invasively detect biochemical changes in the tendons, ligaments, and cartilage that precede structural damage and may offer a great advance in the diagnosis and treatment of these debilitating conditions [25,26,27]. For soft tissue injuries of the palmar aspect of the metacarpal/tarsal regions, MRI is considered more accurate that U/S due to the risk of underdiagnosing injuries [28]. For the diagnosis of distal structure lesions, this may be conducted in a standing manner, with an open U-shaped MRI that is designed to enable imaging up to the level of the carpus and tarsus excluding the need for general anesthesia [29]. However, its high cost prevents its widespread use in initial clinical assessments or during subsequent follow-up examinations to characterize the progression of healing, and the technique requires the availability of professionals trained in interpreting the images obtained.

A CT scan is also a non-invasive procedure that involves the use of radiation, enabling the visualization of bones and soft tissues, producing a 3D image reconstruction [30]. It usually implies general anesthesia, but nowadays, there are also standing CTs. CT arthrography is a very useful diagnostic tool for assessing cartilage defects in metacarpo/tarso-phalangeal joints due to its short acquisition time, its specificity and sensitivity, and it might also be more accurate than MRI. However, MRI allows for a better assessment of the soft tissues and subchondral bone, being a useful technique for joint evaluation [31]. A study was performed comparing CT arthrography (CTA) and MRI sensitivity and specificity in articular cartilage defects in equine metacarpophalangeal joints. The CTA scan sensitivity and specificity were 0.82 and 0.96, respectively, and were significantly higher than those of the MRI (0.41 and 0.93, respectively) in detecting overall cartilage defects (no defect vs. defect). CTA is considered a valuable tool in the diagnosis of articular cartilage defects. CTA specificity is greater than that of MRI, and their sensitivities are very similar, thus concluding that CTA may be preferred, in this case, over MRI due to higher availability and easier image acquisition [26].

Nuclear scintigraphy involves the intravenous administration of a radioisotope that binds to bone structures, reflecting the osteoblast activity and organ function rather than providing detailed anatomical information. This approach enables the diagnosis of skeletal activity and injuries. Nuclear scintigraphy is particularly employed in the evaluation of lameness and poor performance, offering high sensitivity in detecting osseous remodeling associated with stress fractures and stress-related bone injuries [32]. The radioisotope uptake (IRU) by bone might not reflect the pain focus, only bone activity, and can also be associated with other musculoskeletal injuries rather than osseous [33]. Therefore, an in-depth investigation must be conducted, and it is advisable to use this technique together with regional nerve blocks and other diagnostic imaging techniques to better identify the injuries and focus of lameness. Scintigraphy has the potential to provide valuable information, but interpretation is not always straightforward, requiring careful correlation with other clinical and imaging findings [33].

PET scans have recently been added in the diagnostic panorama as a new and valuable tool available for equine musculoskeletal diagnosis, mainly those from the distal limb. Its use is more commonly documented in foot, fetlock, and tarsal injuries [34]. 

PET scan is a non-invasive nuclear medicine imaging technique, functioning as a cross-sectional modality. This entails that, like scintigraphy, a radiotracer is administered to the patient. Unlike the 2D images obtained in scintigraphy, PET scanning captures images in 3D, enabling the creation of multiplanar reconstructions and volume renderings. The tracer predominantly used in equine PET imaging is the radioactive form known as 18F-sodium fluoride (^18^F-NaF) [34]. Utilizing radioactive tracers, equine PET scans provide numerous benefits in diagnosing and managing health issues in horses. These advantages include the ability for early detection and comprehensive assessments as they can pinpoint metabolic alterations at the molecular level before structural changes are visible on other imaging techniques, revealing the extent of disease or injury through detailed imaging, thereby offering deep insights into a range of equine health conditions [34,35]. PET scan images can even be integrated with CT and MRI images in order to more accurately diagnose the injury site, and may now be used in a standing manner with the equine under sedation [34]. 

## 4. Treatment Options

A diverse array of therapeutic modalities is available to address musculoskeletal injuries, ranging from straightforward pain management and inflammation medications to the use of physical rehabilitation with the shared goal of enhancing the biomechanics and restoring function to affected tissues [36]. A relatively recent entrant into the musculoskeletal therapeutic panorama are regenerative therapies, also known as orthobiologics, which assert the ability to regenerate injured tissues or structures. The knowledge and development in this field are progressing rapidly. Both conservative and regenerative therapies play crucial roles in managing musculoskeletal injuries, offering a spectrum of options for horse owners, trainers, and veterinarians to tailor treatments based on the nature and severity of the condition. The choice between these approaches often depends on factors like the specific injury, the horse’s individual characteristics, and the desired outcome for the athlete’s performance and well-being. Subsequently, this discussion will delve into the conservative and regenerative therapeutic options available for treating the aforementioned musculoskeletal pathologies (Table 1).

### 4.1. Conservative Therapies

Conservative therapies have the primary objective of alleviating pain, reducing inflammation, promoting healing, and restoring function. These approaches resort to the use of physical rehabilitation and therapeutic exercises as well as the administration of pharmacological agents. These interventions aim to manage symptoms and promote the gradual recovery of musculoskeletal health without resorting to more invasive procedures [6,15,37]. Conservative therapies include physiotherapeutic, pharmacologic, and surgical methods. Physiotherapeutic techniques include manual therapies, passive stretching, tissue and joint mobilization, thermal therapy, kinesiotape/bandage, and electrotherapy (magnetic field, electrostimulation, therapeutic ultrasound (U/S), laser, extracorporeal shockwave, vibration plates). Pharmacological methods include anti-inflammatories (AIs) (nonsteroidal anti-inflammatories (NSAIDs), corticosteroids (SAIDs)), hyaluronic acid (HyA), polysulfated glycosaminoglycans (PSGAGs), pentosan polysulfate (PPS), polyacrylamide gel, and bisphosphonates to manage pain and inflammation. Surgical techniques include tendon splitting for tendonitis, whose outcome is regenerative.

#### 4.1.1. Physiotherapeutic Modalities

Physical treatment and rehabilitation exercises play a crucial role in promoting the sound musculoskeletal recovery of horses, offering solutions in both preventive and therapeutic scenarios for athletes [36]. In equine sports, physiotherapy primarily targets the soft tissues involved, and is deemed essential to the overall recovery process [36]. 

##### Physical and Mechanical Agents

Manual Therapy

Manual therapy consists of applying the therapist’s hands to the patient’s body for diagnostic or therapeutic purposes. Passive stretching, a modality within manual therapy, is used to improve the range of motion, prevent injury, and decrease pain. It is recommended that each stretch be performed for 3–5 min, once daily, over 3–7 days per week to provide an adequate stimulus [38]. Tissue mobilization implies a deeper mobilization of tissues including massage, myofascial release, tissue mobilization to break down myofascial adhesions such as scar tissue, decrease blood or tissue fluids, and relax muscle tension to optimize fascia, muscle, and tendon function. Joint mobilization implies the active or passive mobilization of joints to increase the range of motion and reduce stiffness and pain. This technique implies small oscillating and gliding movements perpendicular or parallel to the joint axis.

Thermal therapy

Thermal therapy is perhaps the most widely used type of physical therapy and the easiest to apply. It varies from cold to heat therapy; their use is determined by the time elapsed after injury and by the desired objective. The range of thermal modalities is very wide, as demonstrated below [39].

Cold therapy

Cold therapy should be applied in the first 24–48 h after injury. After this, it can be performed in alternance with hot therapy up to 72 h. The main benefits of cold therapy are a decrease in local circulation, tissue swelling, and pain. Reduced blood flow will decrease hemorrhage and the extravasation of inflammatory cells as well as reduce tissue metabolism and enzymatic activity, inhibiting some of the effects of inflammatory mediators, reducing cellular oxygen demands, and decreasing subsequent hypoxic injury. Cold therapy also provides an analgesic effect by decreasing nerve conduction velocity. These benefits are more effective immediately after injury or surgery. After a minimum of 15 min of cold/ice therapy, the tissue temperature decreases 10–15 °C; the average time of cold therapy is 20–30 min [40,41]. Treatments are repeated every 2–4 h during the first 48 h. There are different methods of applying cold therapy—ice water immersion, ice packs, ice bandages, and cryotherapy. Cryotherapy decreases the skin temperature to 4 °C and is a prohibitive treatment at FEI events in and five days prior to the event due to its analgesic and anti-inflammatory effects. Cold therapy can be administered by directing a cold-water hose onto the specific target site. While this approach is practical, it may not be as effective in reducing tissue temperatures compared to immersion in ice water. However, the physical pressure exerted can still be beneficial in addressing edema and promoting the debridement of wounds [39].

The main advantages of this technique include its simplicity of use and ready availability, relatively low cost (except for spas and baths), effectiveness in acute cases, and multiple effects such as analgesia, restricted blood flow as well as reduced tissue metabolism and activity of inflammatory enzymes.

Heat Therapy

Heat therapy is used from 72 h after injury as it increases the metabolic activity in cells, which leads to induced capillary dilation. This increased blood supply leads to increased supplementation of oxygen and nutrition into the area, and enhances removal of cellular waste products such as prostaglandins, decreasing nerve fiber sensitization and pain.

Heating of dense connective tissues increases extensibility and flexibility due to the effects on collagen molecular bonding. The heating and stretching of tissues around joints over a period of time can increase the range of joint movement. Heat also has effects in muscle spasms as it will relax muscles and decrease spasms. In wounds, it will also increase the healing response and improve edema resorption. An increment of 2–4 °C is required to produce these effects [42]. It is contraindicated in cases with infection or neoplasia, where there is hemorrhage, altered skin sensitivity, burns, circulatory problems, or acute inflammation.

Kynesiotape and Bandage

Kynesiotape is an adhesive tape made of cotton with an elasticity of 130–140% of its neutral state [43]. Kynesiotaping (KT) is a technique consisting of the application of an elastic tape to the skin, capable of acting on its mechanoreceptors to generate analgesic, muscular, and circulatory effects. These effects depend on the way the tape is attached to the skin. The elevation of the tissue triggered by the taping (convolutions) decreases the pressure on the nociceptors and increases blood circulation, providing analgesia [44].

The goal is to enhance the activity of mechanoreceptors and proprioceptive receptors within the skin, fascia, ligaments, and joints. The equine skin boasts a significant presence of sensory nerves and receptors intricately linked to hair follicles, characterized by a thinner epidermis compared to human skin. The application of tape is intended to coordinate the interplay between neural and muscular activity, ultimately achieving the modulation or alteration of locomotion and/or range of motion. There is lack of studies in the literature confirming the efficacy of kynesiotape in horses. Nevertheless, in one study, it did not show any significant effect of kynesiotape in range of motion in extension–flexion or stabilization of the back of the trotting horse [43]. Another study demonstrated that the use of therapeutic bandages resulted in a significant reduction in postoperative swelling of tibio-patellofemoral arthroscopy in horses compared with the control group [45]. In [44], the authors demonstrated that KT led to short-term analgesia.

Therapeutic Exercise

Controlled exercise stands as a fundamental and crucial component of every rehabilitation program, complementing and enhancing the regenerative process. The program typically initiates with complete rest, followed by stall rest, and involves a gradual and systematic escalation in exercise intensity [46]. 

Regarding soft tissue injuries, hand walking should start very soon after injury to promote optimal fiber alignment and prevent restrictive adhesions. Exercise is started hand walking 5–10 min once or twice a day, depending on lesion gravity. Ultrasound and lameness evaluation should be made periodically, every 4–6 weeks, and the exercise level may be increased as improvement is achieved [39]. If the ultrasound image deteriorates or lameness grade increases, the exercise plan should be reevaluated. Controlled exercise alone resulted in successful outcomes for 67% to 71% of horses with soft tissue injuries [47]. 

Maturation of tendon and ligament fibers depends on mechanical loading. After the inflammatory phase of healing, controlled stretching is recommended in order to increase collagen synthesis and improve fiber alignment, resulting in a higher tensile strength [48]. These effects are explained by an increase in the collagen content and extracellular matrix (ECM) produced by tenocytes [22]. The promotion of appropriate orientation and the remodeling of collagen into mature, strong, and optimized tissue is ensured by mechanical stress. Controlled exercise during the chronic remodeling phase provides this conversion and improves the mechanical properties of the healed tendon. The quality of the longitudinal fiber pattern has been linked to prognosis for return to work. Collagen that remains unstressed during the proliferative and remodeling phases remains randomly organized and is weaker than stressed collagen. A prolonged immobilization leads to a tendon with reduced content on water and proteoglycan as well as to weaker and random organized collagen fibers with lower tensile strength and failure at lower strains [49]. It also results in tendon atrophy due to lower vascularization and metabolic rate. 

Clinical studies have shown the benefit of early mobilization following tendon repair and the fact that training has improved tensile strength, elastic stiffness, weight, and cross-sectional area of tendons [22,24]. Table 2 presents a suggestion of a controlled exercise program for tendon/ligament injury.

Concerning articular cartilage, slow progressive physical exercise causes significant adaptive changes, there is enlargement of the cells and nuclei of chondrocytes, and an increase in the proteoglycan content and cartilage thickness. Nevertheless, if the exercise is strenuous or misconducted, it may lead to a cartilage degeneration process. The same happens with bone, as bone tissue adapts to weight-bearing and muscular workout by increasing bone mass and density through osteoblast stimulation. This remodeling cycle is slow, taking several months to occur, and the achieved bone mass also depends on genetic, nutritional, and hormonal factors. Immobilization causes the reverse effect on bone tissue, ultimately leading to osteoporosis [48]. Table 3 presents a suggestion for a controlled exercise program for bone injuries. An ideal program is based on individual patient and lesion specificities and requires periodic controlled veterinary check-ups [46]. 

Water Exercise—Hydrotherapy

The most renowned modalities of exercise in water for horses are swimming pools (complete flotation), and water treadmills (WTs) (semi flotation) [50]. 

Marked locomotor differences exist between swimming and exercise on a WT. Usually, when swimming, horses stop forelimb movement, presenting only hindlimb movements. From the rehabilitation point of view, it is interesting and important. They use their forelimbs to maintain balance and hindlimbs for propulsion. Extreme range of motion (ROM) though the hip, stifle, and hock joints are observed in horses during swimming. Moreover, horses adopt a lordotic posture with cervical thoracolumbar and pelvic extension, so caution is recommended when using swimming for horses with thoracolumbar, sacroiliac, hip, stifle, or hock injuries [50]. On WTs, as the water depth increases, the buoyancy increases, the impact shock reduces, and hydrostatic pressure on the limbs increases, all of which have potential benefits for the rehabilitation of certain conditions [51]. Additionally, drag increases, which has the potential to limit limb protraction, alter muscle use, and change stride pattern [51]. A WT exercise session is equivalent to a challenging ground schooling session [51].

Water exercise presents a wide range of advantages: increases joint mobility and its ROM, promotes normal motor patterns, prevents muscle atrophy, increases muscle activation and strength, increases in soft tissue flexibility, reduces edema and joint effusion, reduces comorbidities caused secondarily to primary joint pathology as well as stress applied to the limb, increases joint range of motion, and decreases pain and inflammation [52,53,54]. Limitations of these techniques concern the fact that non-diagnosed injuries may worsen with their overstimulation, and the presence of skin disease or wounds and water mistreatment may lead to cross-infections [55,56]. A good evaluation of each pathology should be carried out before recommending water exercises to understand whether the benefit is real [36]. Regular monitoring of the gait pattern throughout rehabilitation either by a therapist/vet or both is recommended.

##### Electrotherapy

Therapeutic Ultrasound

Therapeutic ultrasound (U/S), an electric device whose action is based on thermal effects, may be used for superficial and/or the deep heating of tissues. Ultrasound selectively heats tissue with high protein/collagen content. The most intense heating occurs at tissue interfaces such as the skin, tendons, and fluid [31,42]. To achieve therapeutic effects, there must be a temperature increase of at least 2 °C in the tendons [42]. In equine epaxial muscles, the mean temperature rise after 20 min of treatment at 3.3 MHz at 1.5 W/cm^2^ was 1.3 °C at a depth of 1.0 cm, 0.7 °C at 4.0 cm, and 0.7 °C at 8 cm. However, temperatures in the tendons were significantly elevated following 10 min of treatment at 3.3 MHz: the mean temperature rise was 3.5 °C in the SDFT and 2.5 °C in the DDFT at the end of the 1.0 W/cm treatment and 5.2 °C in the SDFT and 3.0 °C in the DDFT at the end of the 1.5 W/cm treatment [31]. The other benefit of therapeutic U/S is that sound waves cause a deep massage of tissues known as cavitation. This massage is caused by the expansion and compression of tissues and fluids that enhance tissue healing. For example, fibrous connective tissue scars may be effectively stretched using this technique [42].

This technique is advantageous in articular and tendon disabilities. It addresses joint mobility limitations with the objective of elevating the temperature of connective tissues before engaging in stretching or ROM exercises. Previous observations indicate that warming deep tissues, either before or during stretching, yields a more pronounced impact on tissue length and reduces the risk of injury compared to stretching in isolation. The joint capsule, rich in collagen, frequently contributes to restricting joint motion. Ultrasound energy is efficiently absorbed by collagenous tissue, augmenting its elastic properties [57].

A study found differences among the rates of tissue heating between different tissues. The explanation concerns the thermal and acoustical properties of the tissue through which the continuous sound waves travelled. The difference in the rate of tissue heating among species is likely the result of the distinct acoustic properties of the tissues based on anatomic location and the variation in tissue composition between species [42].

Laser

The term LASER is an acronym for light amplification by stimulated emission of radiation. Laser therapy has been widely used in equine medicine rehabilitation and is gaining much more attention as it is a safe and non-invasive method. It is a device that produces coherent, collimated, and monochromatic light through a process of optical amplification. Laser devices have different classifications—I, II, III, and IV, but only class 3B and IV have therapeutic abilities. Class 3B are therapeutic lasers that have a power output from 5 to 500 mW, and are called low level laser therapy (LLLT). Class IV lasers are therapeutic lasers with a much higher power, above 500 mW, have thermal effects, and are called high intensity laser therapy (HILT) [58]. The use of class IV lasers in FEI competitions is not allowed.

Nowadays, many studies reflect the therapeutic benefits of class LLLT and HILT in tendonitis/desmitis treatment and OA amelioration.

Low-level light therapy generally employs light at the red and near-infrared spectral band (390–1100 nm) to modulate biological activity, without the generation of heat [58,59].

Several studies have presented that LLLT alone can effectively be applied to treat various musculoskeletal disorders [60,61,62,63,64]. LLLT enables tendon healing by promoting angiogenesis under hypoxia, increasing the amount of collagen type III, endorsing the proliferation of fibroblasts, and reducing inflammatory responses. However, it should also be noted that in the final phase of tendon repair, the use of LLLT causes the excessive upregulation of some growth factors, which might lead to tendon fibrosis [62].

LLLT therapeutic efficacy has also been evidenced in joints and articular cartilage, having been demonstrated that beside the anti-inflammatory effects, it promotes a fast recuperation and regeneration of the articular cartilage [65]. In fractures, LLLT increases the bone regeneration process only in the first weeks after the fracture, indicating that LLLT is effective only in the early stages of the process [64]. It is recommended as a physical agent to be used concomitantly with rehabilitation programs [63,66].

Regarding the HILT effects, it presents the same as the LLLT, with additional photothermal effects on soft tissues. The skin temperature rises approximately 3 °C, increasing vascularization and the intensity of metabolic processes in the cells [67,68]. For tendon and ligament injuries, it was demonstrated that it reduced pain, swelling, lameness, and promoted healing, thus reducing the injury percentage, therefore being useful as a supportive therapy for the healing of tendon and ligament injuries in horses [68]. Additionally, HILT seems to be efficient in reducing pain and providing functional improvements in patients with knee OA [69,70,71]. 

Before applying this therapy, patient preparation is essential to ensure that the skin is clean and free of any materials that could absorb light [72].

Extracorporeal Shock Wave

Extracorporeal shock wave therapy (ESWT) is a well-investigated and widely used non-invasive treatment modality for many equine musculoskeletal disorders. Acoustic waves are applied to an injury region that trigger a mechano-transduction cascade. Mechanical energy causes biological effects that lead to an enzymatic response and to the up-regulation of angiogenic growth factors (GFs) responsible for neovascularization as well as the improvement in blood supply and tissue regeneration, thus improving the healing process [73]. 

The application of ESWT in chronic tendinopathies stimulates neovascularization, alleviating pain, and initiating the repair of the chronically inflamed tissues [74]. Its therapeutical value in calcified tendonitis is also largely described [75]. Additionally, ESWT has been shown to improve lameness, decrease the time of healing, and improve the ultrasonographic appearance of tendon and ligament injuries. The optimization of collagen synthesis, maturation, and strength progressively increases the tendon tensile strength and hence recovery. These findings account for the gradual and long-term benefits of shock wave therapy in tendinopathy [76]. 

In knee OA, ESWT has demonstrated clinical benefits for pain and an improvement in physical function. In acute fractures, ESWT treatments enhanced callus formation and induced cortical bone formation. In these cases, the effect of ESWT appeared to be time-dependent [77,78]. 

Limitations of ESWT are the potential pain and minor hematomas. To overcome these limitations, pretreatment with laser therapy is described, the results being a faster and/or better treatment outcome than ESWT without laser pretreatment. Combining ESWT with laser pretreatment leads to synergistic effects and is thus superior to either treatment modality alone [79].

Electromagnetic Field

Electromagnetic field therapy operates on the principle of the electrical generation of magnetic waves. This therapeutic action can be achieved through high-frequency electromagnetic waves, also known as pulsed diathermy, which induce heat production by increasing the temperature by 3–4 °C or by lower-frequency electromagnetic waves, referred to as pulsed electromagnetic frequency (PEMF) therapy, which produces magnetic fields within the tissues without causing heating. For equines, there are several PEMF devices including blankets and wraps with coils and energy-generating battery units built into them [80]. It is indicated for bone fractures, non-union fractures, and in decreasing pain and muscle tension and spasm [80]. Usually, treatment protocols are established by the device manufacturers and are based on the frequency of the pulses and the treatment time.

Electrostimulation

An electrical current is applied to surface electrodes to produce controlled movement of the skin, muscle, tendon, and associated ligaments. Some of the important advantages of electrotherapy are an improved quality of healing and shortened rehabilitation time. Electrotherapy devices can be placed into two categories: sensory nerve or motor nerve stimulators [81]. 

Transcutaneous Electrical Nerve Stimulation (TENS)

TENS provides pain relief through electrical stimulation in the low-frequency range (<250 Hz) by using appropriate pulse durations and intensities to activate the desired nerves. It acts primarily via segmental inhibition through pain gating mechanisms [80]. These rely on the activation of larger diameter fibers in peripheral nerves, which in turn help block nociceptive activity in smaller afferent ones. Secondarily, this stimulation of the peripheral nerves can induce a central release of endogenous opiate-like substances, which can have a descending inhibitory effect on pain. Limitations concern skin irritation. 

The main indications are pain control in acute and chronic musculoskeletal disorders, edema, and wound healing control [80]. In equines, it is mostly described for superficial flexor tendon injuries in order to decrease pain and edema and epaxial muscle pain [82].

Neural Electrical Muscle Stimulation (NEMS)

Neural electrical muscle devices stimulate motor nerves, producing controlled and visible muscle contractions generated by electrical high intensity impulses that are directed toward the target muscle through a surface electrode [83]. 

The main indications are muscle stimulation through α-motor nerve activation and stimulation of de-enervated muscles [80]. Effects such as changes in fiber types and physiological factors of equine muscles, muscle strength and hypertrophy, muscle spasm, and hypertonicity have been described [82].

Vibration Plates

The use of vibration platforms in equine rehabilitation is gaining more support. The main indication is to be used prior to exercise to mimic a warm-up effect caused by vibration, reducing injuries during exercise [84]. It is theorized, although not substantiated, that vibration platforms cause longer stride lengths, lower lameness scores, and higher heart rates after treatment [85]. However, it appears to have an acute relaxation effect in stalled, healthy horses [85,86]. Further studies need to be performed.

#### 4.1.2. Pharmacologic Conservative Therapies

##### Anti-Inflammatories

In musculoskeletal disease, anti-inflammatories—NSAIDs and SAIDs—are the most prescribed and used drugs either administered orally (PO) (both), endovenous (IV) (both), intramuscular (IM) (SAIDs), or injected intra-articularly (IA) (SAIDs). They can relieve pain and reduce inflammation through the inhibition of proinflammatory prostaglandin production by cyclooxygenase enzymes. 

Despite this, as soon as tendonitis or OA is triggered, their histopathology clearly reveals that they have a degenerative course instead of an inflammatory one [87], therefore, anti-inflammatories do not alter the course of the disease, they only relieve symptoms inherent to the pathology [37]. Additionally, these drugs can impair healing by the downregulation of the cycloxigenase-2 (COX-2) pathway for tendon and bone injuries. This entails a profound understanding of the early inflammatory cascade and how it might affect the treatment [88]. Some evidence suggests that NSAIDs may impair the tenogenic differentiation of the mesenchymal stem cells, drawing differentiation toward adipogenic differentiation, and negatively influencing the healing process, thus leading to scar tissue formation and the impairment of functional outcomes [37]. Moreover, the risk of renal, cardiovascular, and gastrointestinal side effects must also be considered.

Corticosteroids and local injections in tendons are not advisable as they may induce tendon fibrosis. Evidence suggests that they are not effective and do not represent any advantage in tendon repair [37,89,90,91].

Regarding the use of intra-articular (IA) SAIDs with OA, these may be associated with moderate improvement in pain and function but with low duration [92]. The beneficial effects of IA SAIDs are rapid in onset, but may be relatively short lived (approximately one to three weeks) [93].

The use of anti-inflammatories must be very well-balanced because if on the one hand, aberrant cellular activity in the inflammatory phase often results in impaired tissue healing and defective host responses with over-fibrosis and scarring; on the other hand, inflammation is part of the regenerative process and recruits a number of immune cell subtypes that have an impact on tissue healing processes [94]. Thus, in acute stages of inflammation, they might be considered based on a short-term use (3–5 days), but their long-term use is not recommended, as inflammation is critical for normal tissue repair, aiding debris clearance and signaling tissue repair [95].

##### Hyaluronic Acid (HyA)

HyA is a non-sulfated glycosaminoglycan (GAGs) and is clinically used for the treatment and medical management of equine acute tendonitis and OA [96,97,98,99,100]. There are several commercial products licensed for injectable use in equine medicine such as Hyalovet^®^ 20 (Boehringer Ingelheim, Milan, Italy) for IA administration, HY-50^®^ (Dechra, Northwich, UK) for IA or IV administration, Hyonate 10 mg/mL (Boehringer Ingelheim, Amesterdam, The Netherlands) for IA or IV administration, Gel-50^®^ (Equimed, Allentown, PA, USA) for IA or IV administration, and Legend^®^ (Boehringer Ingelheim Animal Health, Duluth, GA, USA Inc.) for IA or IV administration.

HyA is considered a safe and cost-effective therapeutic for helping to lower the side effects of OA and is frequently used in clinical routine. HyA provides lubrication to the joints and chondroprotective effects, secondary to an inhibition of the production of nitric oxide, a mediator that enhances cartilage degeneration and chondrocyte death. It also limits the progression of OA lesions by stabilizing proteoglycan structure, limiting the enzymatic breakdown associated with degenerative arthritis [101]. HyA improves the viscosity of synovial fluid (SF), helping its physiological function by acting as a buffer and stabilizer of lubrication and as an anti-inflammatory and analgesic due to increased joint lubrication, resulting in decreased pain in unstable joints [102]. Treatment can be carried out IA, IV, or PO, with IA being more effective. A rest period (12–24 h) is advised after IA treatment [99,103]. The higher the molecular-weight hyaluronic acid, the more efficacious the treatment of OA [97]. Higher molecular weight HyA may provide superior chondroprotective, proteoglycan/glycosaminoglycan synthesis, anti-inflammatory, mechanical, and analgesic effects [104].

In the past, evidence has demonstrated that the IA injection of HyA and SAIDs improved the performance of race horses with traumatic arthritis, and since then, this association has been widely used [105]. However, there is no scientific evidence that hyaluronic acid combined or not with anti-inflammatory drugs is effective in the long run, and that the association with SAIDs is more effective in reducing lameness than HyA itself [102].

Lately, the development of alternative treatments to the classic HyA and corticoids such as platelet rich plasma (PRP) has created the need to compare the treatment effectiveness of these treatment options. Several studies have demonstrated that PRPs in combination with HyA are more effective reducing pain than PRPs or HyA alone [106,107,108,109].

In tendinopathies, HyA also provides analgesia and has been confirmed to be effective on functional improvement as it allows for tendon gliding, reduces adhesions, creates better tendon architectural organization, and limits inflammation [100,110].

Wound healing and immunosuppressive properties have also been reported in in vitro and in vivo studies [96,111]. Its beneficial action to the repair process is stronger after acute tendonitis and should be used soon after injury. 

New products that alter the composition of the HyA molecule are continuously being developed as well as combinations with other drugs to enhance their effects. HyA holds significant potential both as a therapeutic agent on its own and as a scaffold when combined with other therapeutic molecules, and it remains the focus of ongoing research. Nowadays, the association of HyA with mesenchymal stem/stromal cells (MSCs) is being largely studied for the treatment of cartilage repair using HyA as scaffolds for MSC implantation [98,112,113,114,115,116]. 

To sum up, HyA presents analgesic, anti-inflammatory, and lubricative effects, improving organ function [110,111]. 

##### PSGAGs

PSGAGs consist of low-molecular-weight polysulfated glycosaminoglycans (GAGs), ranging approximately from 6000 to 10,000 Da, of animal origin, closely resembling the structure of chondroitin sulfate, which is the predominant GAG found in healthy cartilage [117]. In equine medicine, PSGAG is licensed under the name Adequan^®^ (Luitpold Pharmaceuticals, Inc., Shirley, NY, USA) and is administered via IM. PSGAGs have a long history of demonstrated safety and perceived effectiveness in equine OA prevention, being primarily used to prevent, slow down, and reverse the morphological changes in cartilaginous lesions caused by OA, thus preventing cartilage degeneration [118]. It also presents the ability to reduce inflammation, repair joint cartilage, and promote hyaluronic acid production, thus restoring synovial joint lubrication, alleviating clinical signs, and improving the horse’s quality of life and performance. Its application spans early OA indicators to chronic conditions, serving as a standard treatment approach, also being reported for tendon and ligament injuries [119]. 

##### PPS

Pentosan polysulfate (PPS) is similar to PSGAG but has a vegetal origin. Its molecular structure and function closely align with those of the naturally occurring glycosaminoglycan substances that play a key role in the maintenance and repair of cartilage and connective tissues. PPS exhibits anti-inflammatory, anti-coagulant, and fibrinolytic properties and promotes the synthesis of hyaluronan, making it effective in endorsing cartilage repair, reducing cartilage fibrillation, improving joint function, and alleviating pain associated with OA [117,120]. Some studies have demonstrated that it presents more benefits than PSGAGs when administered IM. Zycosan^®^ (Dechra, Overland Park, KS, USA) is the licensed PPS for equine medicine and is used for the control of clinical signs associated with OA.

##### Polyacrylamide Hydrogel

Polyacrylamide hydrogels (PHyds) are licensed in equine medicine under the brands Arthramid^®^ Vet (Polyacrylamide hydrogel 2.5%, Revatis, Aye, Belgium) and Noltrex^-^vet™ (Polyacrilamide hydrogel 4%, Bioform®, Moscow, Russia).

PHyds have appeared more recently than HyA, and is a 100% synthetic product, are non-soluble, and essentially act as a substitute for SF, increasing joint lubrication and consequently joint pain/inflammation, thus improving joint function. In an in vivo study with rabbits, it was possible to detect the presence of the hydrogel in the joint cartilage space at day 60 after one single dose treatment [121]. The efficacy of PHyd can be possibly explained because its molecular weight is three times greater than HyA, thus preventing the degenerative process caused by the inflammatory cytokines present in the SF of the joints [122]. 

The intra-articular administration route is more efficient than IV or PO, is effective at reducing lameness caused by OA in horses, and has a long period of action enabling their physical activities and increasing the welfare of horses [102].

In a recent study, it was demonstrated that intra-articular 2.5% PHyd is highly effective (82.5% free of lameness horses at 2-year follow-up), lasting, and safe for the treatment of equine OA. No other medical treatment has proven such prolonged efficacy. These studies enhanced the belief that the hydrogel exerts its effects through integration in the synovial membrane, increasing joint elasticity and viscosupplementation, protecting articular surfaces, and preventing pro-inflammatory cytokines from exerting their effects, potentiating OA [122,123]. These studies also suggest the absence of intra-articular neurotoxic effects or fibrosis [123].

All of these studies support the application of a polyacrylamide hydrogel in reducing lameness caused by OA in horses due to its long-lasting viscoelastic supplementation. Its association with other therapies such as PRPs or stem cells could be beneficial.

##### Bisphosphonates: Tiludronate and Clodronate

Bisphosphonates are widely used in both human and equine medicine due to its ability to reduce bone resorption and inhibit osteoclastic activity. In equine medicine, tiludronate was the first bisphosphonate to be approved. It mainly acts as an antiresorptive drug, reducing the ability of the osteoclasts to degrade the bone matrix, although anti-inflammatory and analgesic properties mediated by other mechanisms are also recognized and consensual [124,125,126,127,128,129,130,131,132,133,134,135]. Tiludronate disodium (Tildren^®^, Ceva Animal Health LLC, Lenexa, KS, USA) and clodronate disodium (Osphos^®^, Dechra, Ltd., Staffordshire, UK) are the bisphosphonate drugs that are licensed for use in horses, with its label use on horses older than 4 years old. These two products are non-nitrogen containing bisphosphonates that reduce osteoclastic bone resorption by causing osteoclast apoptosis [136].

Initially, it was mainly used in navicular syndrome and bone spavin.

In navicular syndrome, horses treated with 1 mg/Kg administered via IV injections daily over 10 days for the treatment of navicular disease showed optimal improvement in lameness and return to normal level of activity 2–6 months post treatment [137]. Several studies have proven bisphosphonate effectivity in improving lameness associated with navicular syndrome [129,138,139].

In bone spavin medical treatment, tiludronate, in association with a controlled exercise program, reduces the lameness score and improves radiological images [140,141,142]. 

Nowadays, it is also used for its analgesic action in thoracolumbar spine OA, causing a significant improvement in dorsal flexibility, thus becoming a treatment option for the management of horses with intervertebral lesions and the associated pain [131]. Its use is also valuable to prevent osteopenia in long-term immobilizations [126].

A study conducted in standardbred race horses with fetlock traumatic osteoarticular lesions demonstrated that IV treatment of tiludronate in 500 mL of saline solution decreased the inflammatory process and cartilage degeneration after treatment, meaning that it inhibited the radiographic progression of OA in fetlocks by inhibiting subchondral bone remodeling [128]. The advantage of using tiludronate in young horses to control subchondral bone pain in the initial stages of OA was also highlighted [128]. 

Despite limited data available on its secondary effects, the current literature suggests a good tolerance of tiludronate, with discomfort or colic [126,143,144] and renal damage [134] being the most frequent side effects.

The perspective of the use of bisphosphonates in horses seems bright and growing, as its advantages in osteoclast activity are consolidated. Nevertheless, its usage must be controlled and properly performed.

### 4.2. Surgical Techniques

#### Tendon Splitting

Tendon splitting is a surgical technique performed in acute and chronic tendonitis that has been described for equine tendonitis treatment since the early 70s. 

In acute tendonitis, there is collagen fiber damage and an increase in the cross-section area of the tendon due to intratendinous hemorrhage and inflammatory fluid accumulation within the lesion. In this type of lesion, the objective of this technique is to alleviate pressure from the core lesion, as fluid accumulation within the epitendon and paratendon produce “compartment syndrome”, increasing pressure in the lesion and therefore reducing the perfusion capacity, causing a slow resolution of inflammation and healing. This decompression of the core lesion allows for the evacuation of accumulated inflammatory fluid and promotes vascular ingrowth within the lesion.

In chronic lesions, the procedure is similar, but the objective is to increase vascularization of the scar lesion to promote healing and increase tissue elasticity through the same technique.

Tendon splitting can be conducted blindly, but guided ultrasound is recommended to avoid any damage to healthy tendon fibers and structures other than the injured ones. Briefly, after trichotomy of the area and aseptic preparation of the limb, patient sedation and a high four-point regional nerve block are performed. The ultrasound probe is covered with a sterile lubricant and a sterile sleeve, and allows for tendon visualization.

The stab incision or splitting begins at the most distal aspect of the core lesion to avoid blood contamination of the next stab incision. A #11 scalpel blade is inserted into the medial or lateral surface of the tendon, perpendicular to the ultrasound probe, being its entry and location observed by U/S. The blade is advanced until reaching the lesion, avoiding normal fibers, and is then rotated in upward and downward movements, parallel to the long axis of the tendon. The blade is removed and subsequent stab incisions are made as needed to split the entire length of the core lesion [26]. 

Although this technique was more described in the 70s, 80s, and 90s [27], nowadays, it is still used and was referenced in studies describing the treatment of SL branches with stem cells in race horses, where percutaneous splitting of the ligament was performed in lesions with cross-sectional area (CSA) grade III and IV in order to evacuate the inflammatory fluid within the core lesion, reduce edema, and enhance revascularization, as the reduction in intratendinous swelling through the creation of communication between the core lesion and peritendinous/ligament tissue improves circulation, thus reducing the repair size and enhancing tissue repair organization [27,28]. Splitting the ligament in the higher CSA grades—III and IV—was correlated with a beneficial input in treating these lesions, but more studies need to be performed.

The classification of CSA is in percentage estimates lesion area, as follows: Grade 0, 0%; grade 1, lesion is inferior to 25%; grade 2, lesion represents 25 to 50%; grade 3, lesion represents 50 to 75%; grade 4, lesion is superior to 75% of the cross-sectional area [29].

### 4.3. Regenerative Therapies

The main goal of regenerative medicine is to replace or regenerate cells and tissues, in order to restore the normal structure and function of the injured tissue or organ [145,146].

In contemporary equine orthopedic medicine, there is a growing interest in various regenerative therapeutic approaches, with a notable focus on hemoderivative therapies and stem cell-based therapies. These treatments have gained prominence due to their demonstrated anti-inflammatory effects, immunomodulatory/paracrine properties, regenerative potential, and high tolerability [147].

Hemoderivative therapeutics include PRP [108,148,149,150,151], ACS [152,153,154,155,156], APS [157,158], and α2M [159,160].

Mesenchymal stromal/stem cell-based therapies include cell-based and cell-free therapies. Cell-based therapies resort to the use of stem cells themselves; these are multipotent cells that can be harvested from various tissues. MSCs have the potential to differentiate into different cell types and exert immunomodulatory effects, making them valuable for tissue regeneration. Cell-free therapies rely on cell secreted factors such as cytokines, chemokines, GF, extracellular vesicles (EVs), and exosomes, which present many biological activities as well as therapeutic potential in several organ system and disease contexts. Currently, for equine, the only commercially available MSCs cell-based therapies are under the name of Arti-cell^®^ forte (Boehringer Ingelheim Vetmedica GmbH, Ingelheim am Rhein Germany) and Horstem^®^ (Equicord, Madrid, Spain), and Vet-stem is a laboratory that prepares stem cells from adipose tissue and sells the autologous stem cell product.

The interest in these regenerative approaches stems from their ability to address musculoskeletal injuries at a cellular level, providing a more integrated and potentially more effective treatment strategy. As research in equine regenerative medicine continues to advance, these therapies hold promise for enhancing the overall well-being and performance of horses in diverse disciplines.

#### 4.3.1. Hemoderivatives

Hemoderivatives present anti-inflammatory and healing effects, being used in muscle, tendon, ligament, and joint injuries such as strain injuries, tendonitis, desmitis, osteoarthritis, cartilage injury, and synovitis [160]. They also enable the healing and restoration of function in acute and chronic injuries.

In cases of OA treatment, they represent an advantage when compared with traditional intra-articular treatments (HyA + SAIDs), which are only palliative for pain and inflammation control [161] as they improve clinical signs and appear to be chondrogenic and promote chondrocyte homeostasis [161,162,163]. In cases of tendonitis/desmitis, they also present therapeutic effects, enhancing healing, and leading to the formation of functional tissue without scar formation [164,165].

The common principle across hemoderivatives including PRP, ACS, and APS lies in harnessing the regenerative potential of platelets and their associated bioactive substances to modulate inflammation, support tissue repair, and facilitate healing processes. Each of these approaches offers a personalized autologous solution, utilizing the horse’s own blood components to enhance musculoskeletal health. It is advisable that no NSAID treatments are conducted 1–5 days prior to the preparation of these hemoderivatives [160].

##### PRP

PRP primarily leverages the therapeutic properties of platelets, which play a crucial role in the natural healing response to injury. When tissue damage occurs, platelets become activated and initiate the clotting process, leading to the release of various bioactive substances. The key components released by α granules of the activated platelets include cytokines, growth factors (GFs), and chemokines such as platelet-derived growth factor (PDGF), transforming growth factor-beta (TGF-β), vascular endothelial growth factor (VEGF), and insulin-like growth factor (IGF). These substances are instrumental in modulating the inflammatory response, attracting immune cells to the site of injury. Platelets also contribute to angiogenesis, the formation of new blood vessels, by releasing factors that stimulate the growth and migration of endothelial cells. This process is crucial for supplying oxygen and nutrients to the healing tissue. PRP has garnered significant attention in both equine and human medicine due to its remarkable ability to stimulate the proliferation and migration of fibroblasts, facilitate collagen synthesis, and induce the chemotaxis of macrophages. These cellular processes are crucial for promoting cellular proliferation, tissue healing, and regeneration. PRP has found extensive application in treating musculoskeletal tissue lesions, particularly osteoarthritis (OA) and tendonitis/desmitis, due to its well-established anti-inflammatory and anabolic effects. The proven beneficial effects of PRP underscore its role as a valuable therapeutic tool in promoting tissue repair and regeneration in conditions involving the musculoskeletal system [166,167].

PRP is produced through a centrifugation process of whole blood, during which red blood cells and buffy coat are separated from plasma. Platelets are then aspirated, and a subsequent centrifugation concentrates the platelets in plasma. Platelets release the bioactive factors after degranulation of the α granules in the platelet cytoplasm, which occur upon activation with citrate [168]. Most GFs are released within 1 h of platelet activation and their half-life usually ranges from minutes to hours. This is a simple process that takes approximately 15 min to prepare, being the main device used, a portable centrifuge, which is easy to do in an ambulatory clinic. 

PRP can also be obtained through commercial kits for horses: Restigen PRP^®^ (Zoetis, Lincoln, NE, USA), ACP^™^ (Arthrex GmbH, Munchen, Germany), ACP MAX^™^ (Arthrex GmbH, Munchen, Germany)), Angel PRP^™^ (Arthrex GmbH, Munchen, Germany) or through manual procedures. Although it is described in three different manual protocols, in equine practice, the most widely used protocol involves two centrifugations to concentrate the platelets in a small volume of plasma (e.g., 2–5 mL) for injection in the tendons or intra-articular treatment [169]. PRP can be stored for up to 7 days in cooled storage, however, 24 h is the ideal time of storage at 5 °C because it has been demonstrated that the platelet counting and viability did not change under these conditions [170]. When using a commercial kit, PRP can be aseptically and stably prepared with a consistent platelet content, however, the total platelet count is slightly lower than when using double-centrifugation methods [148].

The platelet content of PRP is affected by several factors such as the breed and age of the horse, the administration of AIs, anticoagulants, blood sampling, and the technical skills of the clinician [171,172]. Depending on the PRP preparation protocol, the cellular and cytokine compositions can vary, with such variability being a main clinical concern as it can potentially influence the therapeutic effects of PRP [173,174]. Nevertheless, all of these present higher levels of TGF-β1, VEGF, and PDGF [148,175].

To sum up, PRP provides a growth factor concentrate that enhances the cellular repair of musculoskeletal lesions [167,174]. Other advantages of PRP as a regenerative therapy are its autologous nature, rapid preparation, and non-invasive collection process.

##### ACS

ACS presents therapeutic effects based on the increase in the interleukin-1 receptor antagonist (IL-1ra) concentration, being therefore known as the interleukin receptor antagonist protein (IRAP). It also presents high concentrations of anti-inflammatory interleukins 4, 10, and 1 (IL-4, IL-10, and IL-1), and growth factors including IGF-1, PDGF, and TGF-β in autologous serum [108,109]. 

In equine medicine, there are commercial kits for the preparation of ACS: Orthokine^®^ vet IRAP (Dechra, Overland Park, KS, USA) and IRAP Pro EAS^®^ (Arthrex, Naples, FL, USA), which is a natural anti-inflammatory product used for the treatment of OA. They have different preparation protocols, but basically consist of whole blood incubation in a syringe containing borosilicate medical glass beads. The blood is then centrifuged to obtain an IL-1ra enhanced serum product that can then be injected intra-articularly or intralesionally. This product may be applied in joint, muscle, and tendon/ligament injuries.

The role of IRAP is very important in OA control as research in molecular biology has discovered that the major inducer of OA is the general inflammatory cytokine interleukin-1β (IL-1β), which plays a key role in accelerating tissue destruction and the repair mechanisms, being one of the major mediators responsible for the pathogenesis of OA as it activates an inflammatory response, leading to cartilage degradation and bone resorption. The proposed mechanism of ACS action is through the blockade of IL-1 receptors, thus inhibiting IL-1 action and preventing the detrimental effects of IL-1β on articular tissues in OA pathophysiology [176,177]. 

Recent studies have also referred to the important contribution of other cytokines such as TGF-β, VEGF, and IGF-1 that would positively influence the treatment response as potent anti-inflammatories and cartilage catabolics [152,162]. IGF-1 is responsible for the stimulation of the production of cartilage matrix components—matrix aggrecan and collagen synthesis—with this profile being another major benefit to add to higher levels of IL-1Ra [178,179].

In tendons, it has been demonstrated that ACS treatment causes an early significant reduction in lameness and leads to a temporary improvement in the ultrasonographic parameters of repair tissue as well as a positive effect on histopathological and biomechanical healing [153,180].

##### APS

APS is an orthobiologic that acts through a combination of cytokines, growth factors, and anti-inflammatory agents, with its main characteristic also being its high concentration of IL-1ra. APS is prepared through an commercially available kit, Prostride^®^ (Zoetis, Lincoln, NE, USA), and the process involves the collection of the horse’s own blood, which is processed with the commercial kit and is intended to stimulate white blood cells (WBC) to produce anti-inflammatory cytokines, concentrating its content in a smaller volume of plasma. This product concentrates IL-1ra, 5.8 times more than in plasma, creating a positive ratio of IL1Ra:IL-1β [162,181]. It is reported to include significantly greater concentrations of IL-1RA, IGF-1, TGF-β, IL-10, and growth factors such as vascular endothelial growth factor (VEGF) and PDGF compared with PRP alone [160].

Its preparation takes 20 min, and then the prepared solution is injected directly into the affected joint or tissue. APS can be prepared using portable centrifugation equipment and is a very simple, quick, and non-invasive technique. The intralesional injections can be performed in a single treatment in an ambulatory-based practice [162,181].

It is designed to reduce inflammation, relieve pain, regenerate tissue, and promote angiogenesis and cell proliferation, capitalizing on the horse’s own biological resources to enhance the healing processes, making it a personalized and potentially effective treatment. 

In horses with naturally occurring OA, APS significantly improved lameness, pain-in-flexion, gait analysis, and range of motion up to 14 days after treatment compared with thee baseline and controls. In equine joint fluid, there was a significant decrease in the protein concentration in treated horses compared to the untreated controls [181]. 

In tendons, it has beneficial effects as an anti-inflammatory and promotes tendon healing [182].

Essentially, the effects of ACS and APS are very similar because they are characterized by higher concentrations of IL-1ra. Nevertheless, the literature has presented some dissimilarities regarding other cytokines, GFs, and anti-inflammatory profiles, attributing some effectiveness variations to these differences [162]. At this point, there is insufficient evidence-based research to support the superiority of APS compared with ACS [162]. However, in the treatment of articular injuries, equine clinicians more widely use IRAP^®^ and Prostride^®^, although there is no evidence to prove that they are more efficacious than PRP in this type of pathology [183,184].

##### α2M

α2M is a broad-spectrum proteinase inhibitor, present in a vertebrate’s plasma, as it binds to proteinases that induce chronic inflammation, especially those released by granulocytes and other inflammatory cells. It has been demonstrated that it can inhibit many cartilage catabolic factors, attenuating post-traumatic OA degeneration. The upregulation of cartilage catabolic factors seems to be a key mechanism for cartilage damage. Therefore, the inhibition of these molecules will prevent disease progression [185].

α2M is naturally present in high levels in plasma and in low levels in SF. It is produced by the liver—being released to plasma—and by chondrocytes and sinoviocytes—being released in SF. In inflammatory events such as OA, the α2M synovial levels do not significantly increase due to its high molecular weight and it does not pass from the plasma to SF, being unable to inhibit severe intra-articular inflammation. Bearing this in mind, several therapies have been developed to administer α2M intra-articularly. It has been proven that this treatment can inhibit inflammation and delay articular cartilage degeneration and bone resorption mediated by the inhibition of catabolic enzymes [185,186,187]. It has also been demonstrated that α2M enhanced the cartilage matrix (i.e., collagen type II and aggrecan synthesis). This fact suggests that α2M may have cartilage repair functions or facilitate the synthesis of cartilage matrix [187]. It has also been suggested that the early administration of α2M may provide cartilage protection by reducing the presence of local catabolic enzymes [187]. In chondrocyte culture, a concentrated α2M serum was found to promote chondrocyte proliferation and reduce apoptosis and catabolic gene expression [145].

Nowadays, to create α2M therapeutic levels within the joint, a process was created that isolates and concentrates α2M from a blood sample. This process was developed and is commercialized as a system—Alpha2EQ^®^ (Astaria Global, Houston, TX, USA). Alfa2EQ^®^ isolates α2M from the horse’s own blood through α active filtration technology, allowing its use as a potent biological anti-inflammatory molecule—α2M—to address equine lameness, joint inflammation, and soft tissue injury. 

To sum up, hemoderivatives represent a new class of regenerative autologous medicinal therapeutics that are evolving rapidly due to their demonstrated efficacy and reduced adverse reactions compared to traditional therapies [188,189]. The production of PRP, ACS, and APS involves the collection of the horse’s own blood, followed by centrifugation and serum collection. In the ACS and APS process, an incubation step before centrifugation is also present. They all exert their actions based on bioactive factors released by platelets, with anti-inflammatory, modulation, and regenerative actions and present different concentrations of specific bioactive factors. α2M also involves the use of the horse’s own blood and its centrifugation, but isolates the α-2 macroglobulin, a multifunctional protein with diverse roles in inflammation, protease inhibition, and immune modulation.

Since they are autologous, they have a personalized nature and avoidance of compatibility issues, thus minimizing the risks of adverse reactions.

However, they also present some limitations. Although safe, promising, and appealing, its use should always require a good evaluation of the patient and should be conducted in a thoughtful way, considering that this is an autologous product, encompassing a considerable inter-individual variability of cytokine and growth factor content, being difficult to assure its constancy and homogeneity [152,172,173,190]. The current literature has failed to identify a preparation method where such variability is limited or negligible [155,189,191,192]. With efficacy differences between the various hemoderivatives, this is not possible yet [162,182,189].

#### 4.3.2. Mesenchymal Stromal/Stem Cell-Based Therapies

Stem cells are undifferentiated cells that can self-renew and differentiate into cells and tissues with specialized functions. Therefore, nowadays, they are the focus for the development of regenerative medicinal therapeutics used to overcome the body’s inability to regenerate damaged tissues after acute or chronic insults. They are classified by their source as embryonic (ESC), adult, and induced pluripotent stem cells (IPSCs) and by their development and differentiation capacity as totipotent, pluripotent, and multipotent cells. Totipotent stem cells are present only in a very early embryo during the morula stage and can develop into all embryonic and extra-embryonic tissues. During early embryonic development, ESC develops and may give rise to all tissue cells in the body, except for extra-embryonic tissues and germ cells. With further development, they gradually lose their pluripotency and become multipotent, which is characterized by the ability to differentiate into limited types of specific cells, often depending on their germ layer origin [193]. Multipotent stem cells might be hematopoietic stem cells (HSCs) or MSCs depending on their origin. HSCs can differentiate into different cells of the immune system, erythrocytes, and platelets, and MSCs into the cells of bone, cartilage, ligaments, tendons, fat, skin, muscle, neural, and connective tissue. Nowadays, there are proposals to change the acronym MSC to “mesenchymal stromal cell”, as these critical advocates suggest that they do not represent true stem cells as there is a lack of some stemness markers [194]. More recently, another nomenclature change was proposed as “medicinal signaling cells”, as these cells home into sites of injury or disease due to the profile of secreted cytokines by these tissues, therefore being signaling cells with medicinal intents [195]. However, recent studies have demonstrated that MSCs can release prostaglandin E2 (PGE2). The autocrine effect of PGE2 displays a major role in the self-renewal ability and immunomodulation of MSCs, thus generating a cascade of events on MSC proliferation, a major characteristic of stem cells, demonstrating MSC stemness [196,197]. 

The International Society for Cellular Therapy proposed a set of standards to define multipotent mesenchymal stromal cells. First, MSCs must be plastic-adherent when maintained in standard culture conditions using tissue culture flasks. Second, ≥95% of the MSC population must express the clusters of differentiation (CD)105, CD73, and CD90, as measured by flow cytometry. Additionally, these cells must lack the expression (≤2% positive) of CD45, CD34, CD14 or CD11b, CD79α or CD19, and human leucocyte antigen (HLA) class II. Third, the cells must be able to follow a trilineage differentiation into osteoblasts, adipocytes, and chondroblasts under standard in vitro differentiating conditions [198].

MSCs exert their function through different paths: homing, that is, migration to the site of injury; differentiation into various cell types that can engraft to the damaged tissue for repair; and secretion of bioactive factors [199]. Initially, it was thought that MSCs migrated to injured tissues, became differentiated, and replaced the local cells. It is currently known that the immunomodulatory capacity of MSC is its main characteristic. This ability is due to the paracrine effect of MSC, the secretion of extracellular vesicles, the immunomodulation of apoptosis, and mitochondrial transfer [199]. 

MSC treatments can be categorized as either autologous or allogeneic, each with its own set of advantages and disadvantages. Opting for autologous treatment offers the advantage of reducing the likelihood of immune reactions, given that the MSCs are derived from the same individual receiving the treatment. However, this approach involves a more time-consuming preparation process including harvesting, processing, and culturing cells from the patient, leading to a delayed treatment onset. Additionally, the individualized production of doses can make autologous treatments more expensive. Furthermore, the patient’s specific characteristics such as sex, age, and health may impact the quality and potency of the MSC treatment. On the other hand, allogeneic treatments, while carrying the risk of possible immune reactions, present the benefit of utilizing cells from a young and healthy donor. This allows for large-scale production and storage in a cell bank, making them readily available for the treatment of acute lesions [200].

No significant differences in efficacy have been established between allogeneic and autologous MSCs for the treatment of musculoskeletal injuries in horses. Therefore, it is suggested that allogeneic MSCs may serve as a safe alternative to autologous MSCs [201]. While autologous MSCs are more commonly used in clinical trials for OA in horses, attributed to their perceived low immunogenicity and lower risk of adverse reactions, recent studies in horses and humans have demonstrated the absence of severe adverse events associated with allogeneic MSCs. This evidence supports the safety of administering allogeneic MSCs [200,202,203,204].

##### Mesenchymal Stromal/Stem Cell Therapies

As previously stated, the use of MSC therapy is one of the potential treatments of orthopedic injuries [3,205,206]. Nowadays, there is proof-of-concept that a variety of tissues have been identified as MSC sources for tissue regeneration and engineering. Bone marrow-MSCs (BM-MSCs) [207,208,209,210], adipose tissue-MSCs (AT-MSCs) [205,207,211,212], synovial membrane-MSCs (SM-MSCs) [200,202,213,214,215,216], amniotic fluid-derived MSCs (AFS-MSCs) [217,218], umbilical cord Wharton jelly’s-MSCs (UC-MCS) [219,220,221,222], periosteum-MSCs (Po-MSCs) [223,224], dental pulp-MSCs (DP-MSCs) [225,226], and muscle tissue-MSCs (MT-MSCs) [227,228,229] are some of these.

Currently, BM-MSCs, AT-MSCs, SM-MSCs, and UC-MSCs are four of the most widely used types of MSCs in the treatment of musculoskeletal lesions.

The literature refers to tendon/ligament injuries with MSCs as very efficacious, suggesting that MSCs can contribute to accelerate and improve the quality of tendon healing by improving the tissue strength, providing a more favorable type I collagen composition, indicating a beneficial therapeutic response to these cells [200,202,230,231,232]. There are several clinical studies using BM-MSCs as the therapeutic option for tendon repair, perhaps because it is the most studied tissue source of MSCs [233,234]. A recent study compared them with UC-MSC, in vitro, and concluded that UC-MSC surpasses other MSCs in its ability to differentiate into tendon-like lineage cells and establish a well-organized tendon-like matrix. In terms of histological properties, UC-MSC promotes a superior regeneration of full-thickness defects when compared to BM- and UCB-MSCs [235]. Notwithstanding, studies with AT-MSCs advocate that this source might be superior regarding their potential to positively influence tendon matrix reorganization and because it is easier to harvest [236,237]. Recently, good results have been achieved by resorting to the use of SM-MSCs, which improved clinical signs and lesion ultrasonographic images, with a return to athletic function and led to no lesion relapse [200,202].

Regarding cartilage defects, BM-MSCs and AT-MSCs have been widely used for the treatment of OA. Each MSC’s tissue origin has its own advantages in cartilage regeneration as they have heterogeneous potential concerning their accessibility, invasion during harvest, immunogenicity, proliferative, chondrogenic, and immunomodulatory abilities [238]. However, as synovium and cartilage have the same origin during the development of synovial joints, SM-MSCs are especially suitable for cartilage and have presented a greater proliferation and chondrogenic ability among other MSCs, suggesting superiority in cartilage repair [209,213,239,240,241,242,243,244,245,246,247,248].

Comparatively with BM-MSCs, SM-MSCs possess a greater colony-forming potential, have a low-density expansion that allows for the retention of multilineage differentiation capacity, and their gene profile matches the chondrocyte and meniscal cell gene profile closer than BM-MSCs [249]. The implantation of MSCs into cartilage defects have shown great promise in both cartilage and subchondral bone repair and regeneration [203,238,246,247,248,250,251,252,253,254,255,256]. 

UC-MSCs present higher proliferation potential, differentiation, and immunogenic abilities from the four most widely used tissues, previously referred to in [257]. They can also release trophic factors that make them an excellent candidate for use in the clinical setting to provide the cell-based restoration of hyaline-like cartilage. Even in allogeneic administrations, these cells stimulate little or no host immune response and can be stored for long periods while maintaining viability [258]. UC-MSCs have also shown the ability for the in vitro induction of the production of glycosaminoglycans and collagen type II [259].

A recent review evidenced significant improvement in pain and function as the main advantages of MSC-based therapy in the treatment of cartilage repair in knees with OA. To sum up, MSCs and the derived exosomes have various functions in the treatment of OA such as an increase in chondrogenesis, proliferation of chondrocytes, reduction in apoptosis, maintenance of autophagy, regulation of synthesis and catabolism of the ECM, regulation of immune response, inhibition of inflammation, and monitoring the mitochondrial dysfunction as MSCs are able to carry out mitochondrial transfer to senescent chondrocytes, thus improving the activity of mitochondrial respiratory chain enzymes and the content of adenosine triphosphates as well as the overall paracrine effect [260].

In skeletal muscle injuries, treatment with AT-MSCs was pointed out as the best choice due to their efficient contribution to myoregeneration. The following characteristics were pointed out as differentiating and advantageous points: their high ex vivo expansion potential and less demanding harvesting than that of BM- or SM-MSCs [261]. Nevertheless, this study refers to autologous treatments. 

Overall, the clinical use of MSCs is safe, is an “easy to do” procedure, and the treatment administration is not very invasive [204].

##### Autologous Chondrocyte Implantation (ACI)

ACI is a novel surgical and regenerative treatment that aims for the regeneration of full-thickness cartilage defects. Chondrocytes are collected from a less loaded area of the joint, digested and expanded, seeded in a scaffold and then injected in the defect region. At the moment, there are several commercial products available such as Cartilife^®^ (Biosolution, Co. Ltd., Seoul, Republic of Korea), MACI^®^ (Vericel Corporation, Sydney, Australia), ChondroCelect^®^ (TiGenix N.V., Leuven, Belgium), Spherox^®^ (CO.DON AG, Teltow, Germany), Chondron™ (CELLONTECH Co. Ltd., Seoul, Republic of Korea), Chondrocytes-T-Ortho-ACI^®^ (Orthocell, Ltd., Murdoch, WA, Australia), and JACC^®^ (Japan Tissue Engineering Co. Ltd., Gamagori, Japan) [262]. Different tissue sources have been used including cartilage, bone marrow, adipose, and umbilical cord tissues to produce chondrocytes. However, those mainly used are autologous bone marrow and cartilage tissues. Recently, there has been a trend shift, with the biggest bet made in allogeneic and adipose tissues [263]. Although these methods can solve the problem of cartilage regeneration to a certain extent, most of the regenerated tissues are fibrous and cartilaginous, which is inferior to hyaline cartilage for the intended purposes of load bearing and joint movement. Therefore, it is difficult to achieve the composition and mechanical properties of natural articular cartilage, and long-term efficacy is not guaranteed [21]. They are relatively successful in relieving pain in patients, but do not result in the regeneration of native tissue [263]. Compared with other techniques such as microfracture or osteochondral autograft/mosaicplasty, ACI seems to be an effective tool for cartilage restoration that may be more efficacious and durable than the other cartilage restoration techniques [264]. Thus, new cell-based and tissue engineering approaches are necessary and continue to be evaluated and optimized with the aim of promoting and inducing cartilage regeneration [263]. 

##### Mesenchymal Stromal/Stem Cell-Free Therapies

As previously discussed, the therapeutic efficacy of MSCs primarily arises from their immunomodulatory function. When exposed to inflammatory stimuli, MSCs secrete a variety of bioactive molecules collectively known as the secretome. The secretome is the collective term for the soluble factors produced by stem cells and employed for their intra- and inter-cell communications [265]. These factors are secreted to the extracellular space, which include soluble factors (cytokines, chemokines, and GFs) as well as non-soluble factors, and extracellular vesicles (EVs) that transport lipids, proteins, ribonucleic acid (RNA), and desoxyribonucleic acid (DNA) subtypes [199,266] (Figure 1). 

EVs can be subdivided into apoptotic bodies, microvesicles, and exosomes. This commixture activates the resident stem cells, and hence mediate the endogenous regeneration. However, the secretome of individual cells and tissues is specific, and changes in response to fluctuations in physiological states or pathological conditions [199]. 

In laboratory settings where MSCs are cultured under specific and adapted conditions, this secretome contributes to the generation of conditioned medium (CM) [267].

Studies have demonstrated that the application of MSC-conditioned medium (MSC-CM) has yielded promising results. This specialized medium, enriched with bioactive factors derived from MSC secretion, has shown efficacy in promoting tissue healing and modulating inflammatory processes due to proangiogenic, antiapoptotic, antifibrotic, anti-inflammatory, and immunomodulatory effects [266]. Previous literature reviews have consistently reported positive outcomes associated with the use of MSC-CM, highlighting its potential as a therapeutic intervention [202,203,267,268]. These findings underscore the importance of exploring MSC-derived secretome and conditioned media as viable treatment options for various inflammatory and tissue injury conditions.

Furthermore, the preservation of the therapeutic action of the parent MSC stands as an additional advantage as each cell type secretes a specific type of bioactive factor [269]. Beyond the biological benefits and addressing the safety concerns associated with the direct application of cells, cell-free therapies offer the potential to avoid immune compatibility, tumorigenicity, and the transmission of infectious diseases potentially related to stem cell therapy as well as several logistical advantages for clinical implementation [265]. These include scalability, ensuring a sufficient supply, and longer shelf-lives [266]. This shift toward cell-free therapies not only enhances the safety considerations, but also streamlines the practical aspects of treatment, making it more accessible, scalable, and feasible for clinical applications.

Certain limitations associated with these treatments pertain to the standardization of MSCs. These include factors such as the age and tissue source of the MSC donor, the duration of MSC preconditioning, the choice of nutritional medium for preculture, the oxygen tension within the culture environment, and the specific preconditioning factors applied [267]. The variability in these factors can influence the characteristics and effectiveness of MSC-derived therapies, emphasizing the need for standardized protocols to enhance consistency and reliability across different treatment approaches. Addressing these considerations will contribute to advancing the field of regenerative medicine and optimizing the therapeutic potential of MSC-based treatments.

Recently, it has been demonstrated that the combination of SM-MSCs and UC-MSC CM could effectively repair a ligament in a reduced time frame, with the achievement of good clinical and imagiological outcomes [202].

A new trend in the regenerative investigation of cell-free therapies is the use of EVs. These have various subtypes and are important mediators in cell-to-cell communication as they carry certain proteins, glycoproteins, lipids, and ribonucleic acids that transmit biological information to support healing in injured tissues [270]. MSC-derived EVs have low-immunogenicity and strong potential for therapeutic applications to treat tissue fibrosis and promote tissue regeneration, and have therefore been proposed as a novel therapeutic agent to mediate immunomodulation and promote regeneration [271]. 

Recent studies have also enhanced some EV characteristics such as their maintenance in systemic circulation and passage through physiological barriers to ultimately exert their effects on recipient cells. Bearing this in mind, they are being studied for different purposes such as regeneration, drug delivery, activity control strategies for pathological EVs, and targeting technologies [272]. However, it is unclear whether using isolated EVs or exosomes excludes an important component of the associated therapeutic effects of cell-based therapy [273]. 

Current investigations support the basis for the clinical translation of MSC exosomes as a cell-free therapy for tissue repair. The literature refers to exosomes as joint protectors against OA damage by promoting cartilage repair, attenuating inflammation, balancing cartilage matrix formation, inhibiting synovitis, and mediating subchondral bone remodeling [274,275]. In tendonitis, they also attenuated the inflammatory phase, increased the proliferation and differentiation of tenocytes, had effects of balancing the tendon extracellular matrix, promoting the tenogenesis of tendon stem cells, and improved enthesis [276,277,278]. In muscular strain and ischemic injuries, exosomes also modulate inflammation, fibrosis, and myogenesis [279,280].

Nevertheless, the use of exosomes is still in its infancy, and approaches for selectively harvesting the exosomes with regenerative potential and screening the regenerative contents have not been achieved yet [276].

Cell-free products can be used naturally or engineered in order to provide superior biocompatibility and biostability, representing a big therapeutical promise in regenerative medicine as they are considered useful for stimulating regeneration with comparable effectiveness to MSCs themselves [168]. These cell-free systems also have the advantage of low immunogenicity, non-cytotoxicity, and non-mutagenicity. In this way, they are becoming a center of interest and are being researched as the best candidates to replace cellular systems in the field of regenerative and immunomodulating medicine [272]. 

## 5. Prognosis

Conservative treatments for OA typically focus on managing joint inflammation and pain, aiming to provide temporary functional improvement. However, they do not halt the progression of the disease, allowing for a continuous degenerative process to unfold.

In the case of tendon and ligament injuries, conservative treatments include NSAIDs, local cooling, and controlled exercise programs. Unfortunately, these approaches frequently result in prolonged and unsuccessful outcomes. Healing in these cases occurs through fibrosis, restricting the return to function and resulting in a loss of tissue elasticity, making the affected area more susceptible to reinjury.

Muscular injuries are commonly addressed with conservative treatments such as NSAIDs, massage, swimming, and other physiotherapeutic modalities. Despite these efforts, the prognosis is often fair, as fibrosis may develop, leading to mechanical lameness and a potential recurrence of the lesion [281]. 

The overall prognosis for musculoskeletal injuries treated conservatively is typically moderate to fair, with clinical signs being alleviated, and when healing occurs, it is through fibrosis. In joint injuries, degeneration continues unabated, and as a result, the affected organ fails to fully recover function, preventing a return to the same performance level.

However, recent advancements in medical therapies have introduced regenerative treatments designed to hamper disease progression, reduce inflammation, and promote tissue regeneration. These innovative therapies mark a significant shift in the clinical paradigm of sports medicine, offering a robust and promising contribution. This transformative approach has the potential to improve the prognosis for musculoskeletal injuries, turning it from moderate to fair to a more optimistic outlook [68].

## 6. Discussion

As emphasized in this work, there is a wide array of therapeutic options for addressing musculoskeletal injuries, with the choice depending on various factors. It is crucial to note that the selection of a specific therapy depends on the type and severity of the musculoskeletal issue as well as the horse’s overall health and intended use.

When lesions occur, there are several conservative and regenerative therapeutic options currently available for managing equine injuries. These treatments aim to promote healing, alleviate pain, and restore functionality. 

Conservative treatments are considered the first line of intervention and aim to alleviate clinical signs, promote healing, and improve overall well-being without resorting to surgery or other measures. They play a crucial role in managing both acute and chronic pain in horses [282]. The suitability of conservative treatments depends on the specific condition, its severity, and the individual needs of the horse. However, their results are usually unsatisfactory, recovery is slow, and lesion relapses are frequent [37].

It is an undeniable fact that conservative treatments are extensively and commonly used in comparison to regenerative treatments [184]. Conservative approaches including physical therapy, medication, and non-invasive interventions tend to be easier to access and are more cost-effective in contrast to certain biologic therapies. The latter often involve advanced technologies, making them more expensive, however, they are becoming more widely used [184]. Cost considerations may arise regarding the perceived effectiveness of conservative treatments when compared to newer or less-studied biologic therapies and have become particularly significant for horse owners and veterinarians working within budget constraints [282]. Nevertheless, while regenerative treatments may initially appear to be more expensive, they often require fewer treatment sessions compared to conventional methods, demonstrate a lower rate of lesion recurrence, and promote better clinical and functional outcomes, often achieving complete regeneration of the affected tissue [283]. Overall, they are more advantageous, and their perceived outcome is increasing as their use is becoming more common [119].

Conservative treatments often benefit from a long history of use and are supported by a substantial body of empirical evidence. When these treatments prove ineffective in halting clinical signs and pathological traits, regenerative treatments emerge as the most effective means, offering both anti-inflammatory and regenerative effects [169,198,199,200]. Nowadays, they represent the most promising class of therapeutics and continue to be in constant development. They are biologic and therefore “drug-free”, having no concerns with clearance, making it very appealing for use in high-level sports. 

Within this class, MSC-based therapies exhibit clinical efficacy, inducing favorable outcomes [193,260,277]. The ability to manufacture or engineer MSCs and their products according to specific pathologies enhances the therapeutic responses based on tissue source, secretome, or cytokine manipulation [199,266,284].

Concerns regarding the immunogenicity and tumorigenicity of stem cell therapies have been mitigated, as no severe adverse reactions have been reported in clinical experiments [197,200,202,203,285]. The use of allogeneic bank cells from healthy donors facilitates prompt treatment in the acute phase, circumventing constraints associated with autologous treatments such as individual as well as the heterogenic and time-consuming processes of harvest and production that may lead to variations to the product’s cell and cytokine composition [286]. All of these factors, which can potentially affect product variability, are a main clinical concern once they can negatively influence the therapeutic effects [287]. Therefore, allogeneic treatments seem to be more advantageous as they skip autologous related limitations.

Among the regenerative therapies, while hemoderivative products have been gaining significant expression in the market with optimistic outcomes, MSC-based products are making strides in that direction. Considerable research has been conducted in this field [288], but there remains a lack of qualitative and quantitative evidence-based data supporting MSC-based regenerative therapies in clinical use, since most of the studies are observational, therefore being dissimilar from each other [4,5,128,138,173,181,203,289,290,291,292,293,294]. The high heterogeneity of the reviewed studies did not allow for a meta-analysis to compare the results between treatments. However, the results have been unanimous, concluding on the effectiveness and achievement of tissue regeneration, which is the ultimate objective of any musculoskeletal treatment [200,202,284,288,295,296]. In fact, in the last 15 years, there has been a shift in the trend among equine practitioners from OA conservative treatments to regenerative treatments [119].

Significant steps are being taken in the pursuit of standardized protocols for the therapeutic production, storage, and application of regenerative treatments. Establishing accurate therapeutic protocols, identifying optimal hemoderivative or MSC-tissue sources for specific diseases, determining suitable dosages, and establishing the ideal intervals between applications for various pathologies are in the spotlight and warrant further exploration [173,297]. In fact, the expanding market launch of regenerative commercial products is making their utilization more attractive and widespread, with the products exhibiting uniform characteristics, whether through devices that prepare them or as final products available on the market.

Additionally, it is essential to delineate effective physiotherapeutic protocols, employing one or more methods, during rehabilitation period and in routine exercise plans [36,39]. A profound understanding of each physiotherapeutic technique is vital, considering this is a crucial aspect of any musculoskeletal treatment and must be used synergistically with other therapies in order to offer a more favorable return to function and improved clinical outcomes [36].

Recognizing the evolving understanding of a multidisciplinary approach is pivotal for achieving the optimal therapeutic results. Ensuring a well-balanced environment is decisive for the health and performance of the equine. A collective effort of the horse-environment is compelling in preventing and managing injuries effectively, fostering a holistic and informed approach.

## 7. Conclusions, Challenges, and Future Research Directions

Currently, a diverse array of musculoskeletal treatments is available. While conservative treatments have their merits, they also come with several limitations. This has propelled regenerative treatments into a position of high importance and hope within the field. 

The regenerative approaches have demonstrated their value and effectiveness, holding value and promise for addressing musculoskeletal issues in novel and potentially more effective ways. They present anti-inflammatory abilities, enable a return to function and tissue sustainability due to their regenerative competence in a faster time frame, providing a beacon of optimism for both practitioners and the horse community. It is important to bear in mind that the synergistic integration of physiotherapeutic techniques, together with good horsemanship practices and regenerative approaches, holds the promise of delivering superior outcomes.

Promising results in the realm of tissue repair and regeneration are evident in clinical studies with horses. At present, considering MSC-based therapies as the most capable of facing musculoskeletal injuries, the primary hurdles concern the strength of the evidence, which is currently hampered by the absence of controlled clinical trials, in order to comprehensively understand the advantages and limitations of each therapy. However, this is a field that is rapidly expanding and experiencing significant growth, propelled by the auspicious results that are currently emerging. Consequently, it is anticipated that this will soon cease to be a limitation. The future will dictate the identification of the most effective tissue source, to correspond it to a specific type of lesion, to design orthobiologics that are readily accessible, easier to administer, that carry minimal risk, and are financially feasible. Further research is imperative to validate the efficacy and establish precise guidelines for clinical implementation. Despite the ongoing quest for answers in regenerative therapies, it seems that they have presently emerged as the most effective approach to address musculoskeletal injuries.

The outlook in regenerative medicine is promising, generating high expectations and capturing extensive attention in both equine and human medicine as they accelerate recovery, promote regeneration, and organ functionality, and therefore, a return to peak performance and quality of life.

## Figures and Tables

**Figure 1 vetsci-11-00190-f001:**
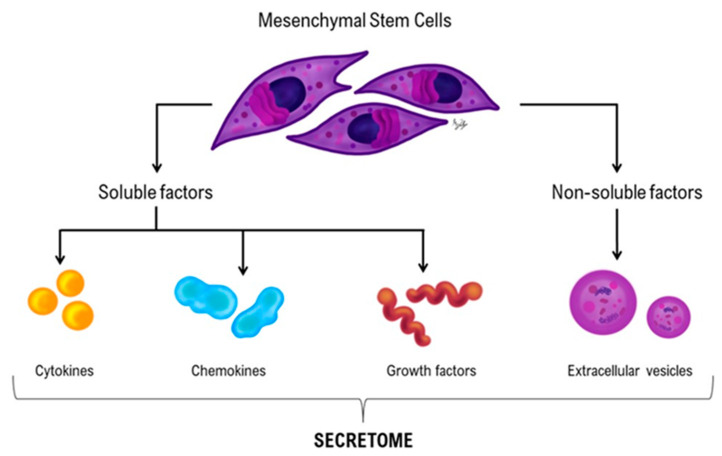
Secretome constitution. The secretome encompasses a varied collection of soluble and non-soluble factors. Soluble factors include cytokines, chemokines, and growth factors. Non-soluble factors include extracellular vesicles (exosomes, apoptotic bodies, and microvesicles) [223].

**Table 1 vetsci-11-00190-t001:** Therapeutical options for musculoskeletal injuries. Therapeutic ultrasound (U/S), anti-inflammatories (AIs), hyaluronic acid (HyA), polysulfated glycosaminoglycans (PSGAGs), pentosan polysulfate (PPS), platelet-rich plasma (PRP), autologous conditioned serum (ACS), autologous protein solution (APS), alfa-2-macroglobulin (α2M), autologous chondrocyte implantation (ACI).

Musculoskeletal Injuries Therapeutical Options				
Conservative Treatments		Surgical Techniques	Regenerative Treatments	
Physiotherapy	Pharmacological		Hemoderivatives	Stem-Cell Based Therapies
Manual Therapy	AI’s	Tendon splitting	PRP	Stem-cell therapy
Thermal therapy	HyA		ACS	ACI
Kynesiotape	PSGAGs		APS	Stem-cell-free therapy
Therapeutic exercise	Pentosan PS		α2M	
Water exercise	Polyacrylamide hydrogel			
Therapeutic U/S	Biphosphonates			
Laser				
Extracorporeal Shockwaves				
Electromagnetic field				
Electrostimulation				
Vibration Plates				

**Table 2 vetsci-11-00190-t002:** Controlled exercise protocol for tendon/ligament injury. The horse is confined to a stall or equivalent size paddock. Adapted from [39].

Weeks after Injury	Exercise	Confinment
**0–4**	Hand walk, 5–10 min, Twice daily.	Stall rest
**5–8**	Hand walk, 10–15 min, Three times daily.	Stall rest or small paddock
**9–12**	Increase time walk 5 min/day Three times daily	Stall rest or small paddock
**13–16**	If sound and continued improvement in lesion parameters: ride at the walk 20–25 min daily, hand walk 30 min daily.	Stall rest or small paddock
**17–20**	Ride at the walk 30 min, add 3–5 min trot. On week 18, add 3–5 min additional trot per week.	Stall rest or small paddock
**21–recovery**	Ride at the walk 30 min, ride at the trot 15 min per session, add 3 min canter. On week 22–24, add 3–5 min canter per session	Small paddock

**Table 3 vetsci-11-00190-t003:** Controlled exercise protocol for bone injury. In the first month, the horse must be confined to stall rest and then start gradually and increasing exercise. Adapted from [46].

Weeks after Injury	Exercise	Confinment
**0–4**	-	Stall rest
**5–6**	Hand walk, 15 min/day.	Stall rest or small paddock
**7–8**	Hand walk, 30 min/day.	Stall rest or small paddock
**9–16**	Exercise in small paddock 6 × 6 m.	Stall rest or small paddock
**16–recovery**	Gradually increase exercise until full work.	Stall rest or small paddock

## Data Availability

The original contributions presented in the study are included in the article, further inquiries can be directed to the corresponding author.

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
