# Peer review of "Equine Musculoskeletal Pathologies: Clinical Approaches and Therapeutical Perspectives—A Review"

_vetsci, 2024, doi:10.3390/vetsci11050190_

Round 1
Reviewer 1 Report
Comments and Suggestions for Authors
The manuscript “Equine musculoskeletal pathologies: clinical approaches and therapeutical perspectives - a review." focus an interesting and very common subject related to equine practice. The review compiles several therapeutic possibilities which may be important to clinicians.
In my opinion, the subject is complex and the manuscript seems too large. Maybe should be considered to divide in Part 1 and Part 2?
In filiation, numbers 6 and 7 are not attributed to anyone?
The simple summary is adequate.
The Abstract is correct and elucidates the content of the manuscript.
The Introduction is ok.
Complementary Diagnostic Exams
Lines 209-210 or line 216 – “perineural nerve blocks,” and intra-articular anaesthesia?
Treatment Options
Table 2 should appear after line 484.
Line 806 – “navicular disease. showed” remove the .
The discussion seems correct.
The conclusions are consistent with the evidence and arguments presented.
Author Response
Porto, 06 of April 2024
Dear Editors
Dear Reviewers
Veterinary Sciences
Title: Equine musculoskeletal pathologies: clinical approaches and therapeutical perspectives - a review. Authors: I.L. Reis, B. Lopes, P. Sousa, A.C. Sousa, A.R. Caseiro, C.M. Mendonça, J.M. Santos, L.M. Atayde, R.D. Alvites, A.C. Maurício.
We are sending the revised form of the manuscript entitled “Equine musculoskeletal pathologies: clinical approaches and therapeutical perspectives - a review”. The authors addressed, point by point, the reviewer’s questions. The changes are shown highlighted in yellow throughout the manuscript. We hope that this revised version of the manuscript fulfils all the requirements in order to be published in the Veterinary Sciences journal.
On behalf of all authors,
Ana Colette Maurício
Professora Catedrática/ Full Professor
Diretora do Departamento de Clínicas Veterinárias
Diretora do Doutoramento em Ciências Veterinárias
ICBAS - Universidade do Porto
Rua Jorge Viterbo Ferreira, nº 228
4050-313 Porto, Portugal
TEL: +351 220 428 000; TELM: +351 919071286
EXT: 5367 (Gab), 5371 (Lab)
ana.colette@hotmail.com; acmauricio@icbas.up.pt
Reviewer #1
The manuscript “Equine musculoskeletal pathologies: clinical approaches and therapeutical perspectives – a review.” Focus an interesting and very common subject related to equine practice.
The review compiles several therapeutic possibilities which may be important to clinicians.
1. In my opinion, the subject is complex and the manuscript seems too large. Maybe should be considered to divide in Part 1 and Part 2?
Thank you very much for your comment.
In fact, the work is very long and although it is not formally divided into 2 parts, it ends up being, with subdivisions 6.1, 6.2 and 6.3 (surgical treatments are now within a new sub-section).
In filiation, numbers 6 and 7 are not attributed to anyone?
In fact, the filiation 6 and 7 were not correct. It was corrected and highlighted in the revised manuscript.
- Lines 209-210 or line 216 – “perineural nerve blocks,” and intraarticular anaesthesia?
Thank you for your pertinent comment. It is already addressed in lines 208 and 215.
“CD tools encompass a broad spectrum, including flexion tests, perineural nerve blocks, intra-articular anaesthesia and various imaging techniques.”
“Perineural nerve blocks and intra-articular anaesthesia aid in the determination of pain localization area [36].”
- Table 2 should appear after line 484.
Thank you for your review. We noticed that when formatting, table 3 disappeared, so table 2 was inserted after line 467 and table 3 after line 481.
- Line 806 – “navicular disease. showed” remove the.
Corrected.
Line 811: the treatment of navicular disease showed optimal improvement.

Reviewer 2 Report
Comments and Suggestions for Authors
5. Complementary Diagnostic Exams
The authors have not distinguished Positron emission tomography (PET) scan as a “Complementary Diagnostic Exams” method. PET scanning has been effectively introduced into equine veterinary medicine, revolutionizing diagnostic capabilities for horses. By utilizing radioactive tracers, PET scans can detect metabolic changes at a cellular level, offering insights into various equine conditions such as lameness, tumors, and neurological disorders. This non-invasive imaging technique provides veterinarians with valuable information to accurately diagnose and treat equine ailments, leading to improved outcomes and enhanced equine healthcare.
Table 1.
The authors have not distinguished equine chiropractic care as a “Physiotherapy” option. Chiropractic care has been effectively introduced into equine veterinary medicine, offering a holistic approach to maintaining the health and performance of horses. By focusing on the alignment and function of the musculoskeletal system, chiropractic techniques help alleviate pain, improve mobility, and enhance overall well-being in equine patients. Integrating chiropractic care into veterinary practice has shown promising results in treating various conditions, such as back pain, muscle tension, and joint dysfunction, contributing to horses' comprehensive care and rehabilitation.
The authors have not distinguished Polysulfated Glycosaminoglycans (PSGAGs) as a “Pharmacological” option. PSGAG has gained recognition as an effective treatment in equine veterinary medicine for managing joint disorders and osteoarthritis in horses. As an injectable medication, PSGAG promotes cartilage repair, reduces inflammation, and preserves joint function. Its ability to enhance joint health and mobility has made it a valuable therapeutic option for equine athletes and aging horses. With proven efficacy and safety profiles, PSGAG has become a cornerstone in treating equine joint ailments, improving the quality of life and horse performance outcomes.
Page 11 of 50
Table 3. There is no companying Table shown.
6.1.2.2 HyA
Legend, Adequan, and Zycosan are not hyaluronic acids used for treatment in equine veterinary medicine; instead, they are different classes of medications to manage joint issues in horses. Legend, containing hyaluronate sodium, is an intravenous injectable used to treat joint dysfunction and synovitis. Adequan, on the other hand, is PSGAG administered via intramuscular injection, aiding in the repair and maintenance of joint cartilage. Zycosan is a proprietary blend of ingredients designed to reduce inflammation and promote joint health, distinct from hyaluronic acid-based treatments. Each of these medications plays a crucial role in the comprehensive management of equine joint conditions, offering veterinarians various therapeutic options tailored to individual patient needs.
6.1.2.4
Page 18 of 50
Please clarify “A study conducted in standardbred race horses…”
Abbreviations
Please closely review all abbreviations; several are missing (e.g., Stem Cells, Synovial Fluid, etc.) and others are misidentified (e.g., AINEs—non-steroidal anti-inflammatory, COX—Cycloxigenase, IGF—insulin growth factor, etc.)
Comments on the Quality of English Language
none
Author Response
Porto, 06 of April 2024
Dear Editors
Dear Reviewers
Veterinary Sciences
Title: Equine musculoskeletal pathologies: clinical approaches and therapeutical perspectives - a review. Authors: I.L. Reis, B. Lopes, P. Sousa, A.C. Sousa, A.R. Caseiro, C.M. Mendonça, J.M. Santos, L.M. Atayde, R.D. Alvites, A.C. Maurício.
We are sending the revised form of the manuscript entitled “Equine musculoskeletal pathologies: clinical approaches and therapeutical perspectives - a review”. The authors addressed, point by point, the reviewer’s questions. The changes are shown highlighted in yellow throughout the manuscript. We hope that this revised version of the manuscript fulfils all the requirements in order to be published in the Veterinary Sciences journal.
On behalf of all authors,
Ana Colette Maurício
Professora Catedrática/ Full Professor
Diretora do Departamento de Clínicas Veterinárias
Diretora do Doutoramento em Ciências Veterinárias
ICBAS - Universidade do Porto
Rua Jorge Viterbo Ferreira, nº 228
4050-313 Porto, Portugal
TEL: +351 220 428 000; TELM: +351 919071286
EXT: 5367 (Gab), 5371 (Lab)
ana.colette@hotmail.com; acmauricio@icbas.up.pt
Reviewer #2
- The authors have not distinguished Positron emission tomography (PET) scan as a “Complementary Diagnostic Exams” method. PET scanning has been effectively introduced
into equine veterinary medicine, revolutionizing diagnostic capabilities for horses. By utilizing radioactive tracers, PET scans can detect metabolic changes at a cellular level, offering insights
into various equine conditions such as lameness, tumors, and neurological disorders. This non-invasive imaging technique provides veterinarians with valuable information to accurately diagnose and treat equine ailments, leading to improved outcomes and enhanced equine healthcare.
Thank you very much for your comment. This is very innovative and its introduction in the market is very recent. It will be a good income in this work as I believe that in Europe, it is not widely present.
Lines 302-319.
“PET scan has been recently added in the diagnostic panorama as a new and valuable tool available for equine musculoskeletal diagnosis, mainly those from distal limb. Its use is more commonly documented in foot, fetlock and tarsal injuries [57].”
“PET scan is a non-invasive nuclear medicine imaging technique, functioning as a cross-sectional modality. This entails that, like scintigraphy, a radiotracer is administered to the patient. Unlike the 2D images obtained in scintigraphy, PET scan captures images in 3D, enabling the creation of multiplanar reconstructions and volume renderings. The tracer predominantly used in equine PET imaging is the radioactive form known as 18F-sodium fluoride (18F-NaF) [57]. Utilizing radioactive tracers, equine PET scans provide numerous benefits in diagnosing and managing health issues in horses. These advantages include the ability for early detection and comprehensive assessments, as they can pinpoint metabolic alterations at the molecular level before structural changes are visible on other imaging techniques, revealing the extent of disease or injury, through detailed imaging thereby offering deep insights into a range of equine health conditions [57,58]. PET scan images can even be integrated with CT and MRI images in order to more accurately diagnose the injury site. It may now be used in a standing manner with the equine under sedation [57].”
- Table 1
The authors have not distinguished equine chiropractic care as a “Physiotherapy” option. Chiropractic care has been effectively introduced into equine veterinary medicine, offering a holistic approach to maintaining the health and performance of horses. By focusing on the alignment and function of the musculoskeletal system, chiropractic techniques help alleviate pain, improve mobility, and enhance overall well-being in equine patients.
Integrating chiropractic care into veterinary practice has shown promising results in treating various conditions, such as back pain, muscle tension, and joint dysfunction, contributing to
horses' comprehensive care and rehabilitation.
Thank you for your comment.
Although not with the explicit name of chiropractic, this type of modality is included in the big group of “Manual Therapy” as chiropractic is a manual therapy with the principles explained in this subsection.
- The authors have not distinguished Polysulfated Glycosaminoglycans (PSGAGs) as a “Pharmacological” option. PSGAG has gained recognition as an effective treatment in equine veterinary medicine for managing joint disorders and osteoarthritis in horses. As an injectable medication, PSGAG promotes cartilage repair, reduces inflammation, and preserves joint function. Its ability to enhance joint health and mobility has made it a valuable therapeutic option for equine athletes and aging horses. With proven efficacy and safety profiles, PSGAG has become a cornerstone in treating equine joint ailments, improving the quality of life and horse performance outcomes.
Thank you for your comment.
In fact, subsection 6.1.2.2. should be “viscosupplementation” and include all of the medicines that sharing the same effect present different characteristics.
Your comment was very pertinent and PSGAGs and Pentosan are now addressed in this work from lines 744-757 and lines 758-767.
“6.1.2.3. PSGAGs
PSGAGs consist of low molecular weight polysulphated glycosaminoglycans (GAG), ranging approximately from 6,000 to 10,000 Da, closely resembling the structure of chondroitin sulphate, which is the predominant GAG found in healthy cartilage [142]. In equine medicine PSGAG is licensed under the name of Adequan® (Luitpold Pharmaceuticals, Inc., Shirley, NY) and is administered via IM. PSGAGs have a long history of demonstrated safety and perceived effectiveness in equine OA prevention being primarily used to prevent, slow down, and reverse the morphological changes in cartilaginous lesions caused by OA, thus preventing cartilage degeneration[143]. It also presents the ability of reducing inflammation, repairing joint cartilage, promoting hyaluronic acid production thus restoring synovial joints lubrication, alleviating clinical signs and improving horse’s quality of life and performance. Its application spans early OA indicators to chronic conditions, serving as a standard treatment approach, being also reported for tendon and ligament injuries [144].”
“6.1.2.4. PPS
Pentosan polysulphate (PPS) is similar to PSGAG but has a vegetal origin. Its molecular structure and function closely align with those of the naturally occurring glycosaminoglycans substances that play a key role in the maintenance and repair of cartilage and connective tissues. PPS exhibits anti-inflammatory, anti-coagulant and fibrinolytic properties, promotes the synthesis of hyaluronan, making it effective in endorsing cartilage repair, reducing cartilage fibrillation, improving joint function, and alleviating pain associated with OA [142,145]. Some studies demonstrate it presents more benefits than PSGAGs, when administered IM. Zycosan® (Dechra, USA), is the licensed PPS for equine medicine, used for the control of clinical signs associated with OA.”
- Page 11 of 51
Table 3. There is no companying Table shown.
Thank you for your comment. It is already addressed.
- HyA
Legend, Adequan, and Zycosan are not hyaluronic acids used for treatment in equine veterinary medicine; instead, they are different classes of medications to manage joint issues in horses.
Legend, containing hyaluronate sodium, is an intravenous injectable used to treat joint dysfunction and synovitis. Adequan, on the other hand, is PSGAG administered via intramuscular injection, aiding in the repair and maintenance of joint cartilage.
Zycosan is a proprietary blend of ingredients designed to reduce inflammation and promote joint health, distinct from hyaluronic acid-based treatments. Each of these medications plays a crucial role in the comprehensive management of equine joint conditions, offering veterinarians various therapeutic options tailored to individual patient needs.
Thank you for your comment. As mentioned before, new paragraphs for PSGAG and PPS were added to correct this part. It was also added in Table 1.
Regarding Legend and the other referred sodium hyaluronates, as the reviewer is aware, hyaluronic acid, can exist in the following forms depending upon the chemical environment in which it is found: as the acid, hyaluronic acid; as the sodium salt, sodium hyaluronate (hyaluronate sodium); or as the hyaluronate anion. These terms may be used interchangeably. In fact, in Legend’s Boehringer page, they use both nominations (Legend (hyaluronate sodium) for Horses | Boehringer Ingelheim Animal Health (bi-animalhealth.com)).
We have considered them as therapeutical options as they ameliorate clinical signs, reducing lameness as they are chondroprotectors and improve synovial joint lubrication.
Line 696: “HyA is a non-sulphated glycosaminoglycan (GAGs) and is clinically used for the treatment and medical management of equine acute tendonitis and OA [120-124].”
- Page 18 of 50
Please clarify “A study conducted in standardbred race horses…”
Thank you for your comment. Corrected.
Line 817 – “A study conducted in standardbred race horses…”
- Abbreviations
Please closely review all abbreviations; several are missing (e.g., Stem Cells, Synovial Fluid, etc.) and others are misidentified (e.g., AINEs—non-steroidal anti-inflammatory, COX—Cycloxigenase,
IGF—insulin growth factor, etc.)
Thank you for your comment. It was corrected in the revised manuscript.

Reviewer 3 Report
Comments and Suggestions for Authors
The manuscript has an interesting theme and objective, but the authors were too ambitious in wanting to cover all the existing literature on musculoskeletal problems, as I will mention later.
The manuscript is excessively long, repetitive and contains absolutely unnecessary information, often losing focus on the initially stipulated objective. I recall that the authors mention in the abstract and simple summary that the aim is "The to compile current therapeutic options for managing these injuries". Something that is lost when they talk about diagnosis, doping, the principles of some therapies (laser, for example).
The whole manuscript will have to be revised, with major rewording to make it more objective and straightforward for readers. It has to stick only to the objective initially proposed.
Introduction
Figure 1 is too basic and unnecessary.
All the information contained in the headings: 2) epidemiology; 3)most common musculoskeletal pathologies; 4) clinical examination; 5)complementary diagnostic exams - is not directed towards the objective of the review. At no point, in this vast expanse of text, do they address treatments. I therefore strongly urge the authors to consider deleting all the information contained here and, after a brief introduction, to start describing the treatment options.
Table 1: In my opinion, and generally in the medical literature, surgery is not considered a conservative treatment, but an interventional one. Consider dividing the table into 3 columns with conservative treatment, surgical treatment and regenerative treatment. In the table's legend, it is necessary to name the acronyms present, for example AI's; HyA; PRP; etc.
Lines 318-330: this information repeats literally what is present in the table. Delete
Lines 346-352: Again, this information is redundant with the rest of the manuscript. If you're going to explore each of these topics ahead, there's no need for more text on them. The manuscript is too big to repeat information unnecessarily
Lines 353-358: This information is already contained in lines 332-336, since Physiotherapeutic modalities are part of Conservative therapies
Lines 375-377: Already mentioned below. Delete
Line 378 and line 401: cold therapy and heat therapy are not numbered separately as a subtitle. So either the authors number them or they should appear in running text and not separated into sections.
Line 407: "Heating prior to exercise...." this isn't therapy, it's injury prevention. This is done on any horse with or without pathology. Prevention and treatment are different matters. The aim here is to summarize the role of each therapy in the treatment of various pathologies and not their role in preventing disease. The authors have to decide whether they want to talk about the preventive role or the therapeutic role. Addressing both topics makes the text too long. This kind of mixing of concepts (treatment vs prevention) occurs throughout the text including discussion. Please revise the manuscript accordingly and eliminate these misconceptions
Lines 437-443: once again, the authors are talking about diagnosis which is not the focus of the work
Line 485: table 3 is not present
Each treatment section has too much text/information. Giving the example of the laser, it is not the aim of this work to explain what the laser is or its principles. There is specialized literature on this. We want to know in which pathologies it is applicable, what effects it causes (and it is possible to summarize this part, since it is not necessary to explain the entire molecular perspective of its effects...if it reduces inflammation it is not necessary to address each component of the inflammatory cascade...clinically it is not interesting to go into so much detail) and how it should be used (protocols for example).
Lines 813-834: it's a huge paragraph! It talks about diagnosis between lines 820-823 and doping in lines 825-834. There's no emphasis on the therapeutic component...
Once again, I recommend a thorough review of the manuscript to eliminate what doesn't belong and to eliminate repetitions of information.
Line 848: "6.1.3 Conservative surgical technique" - in my opinion "conservative surgical.." is a contradiction.
Discussion
Once again, the lack of synthesis is obvious. The text between lines 1355-1373 has no relevance, it talks about prevention and even orthopedic management of foals... This is a great mixture of information that confuses and makes the text difficult to read.
Before evaluating the rest of the discussion in detail, I would ask the authors to delete the paragraphs/sections mentioned by the reviewer and to review the entire manuscript in order to eliminate what is not relevant and, above all, to synthesize the information contained.
I understand that clinically there is an interest in approaching everything in an integrated way, but scientifically it is not correct when there is a pre-defined scientific objective.
Please check the references (references number 25)
Author Response
Porto, 06 of April 2024
Dear Editors
Dear Reviewers
Veterinary Sciences
Title: Equine musculoskeletal pathologies: clinical approaches and therapeutical perspectives - a review. Authors: I.L. Reis, B. Lopes, P. Sousa, A.C. Sousa, A.R. Caseiro, C.M. Mendonça, J.M. Santos, L.M. Atayde, R.D. Alvites, A.C. Maurício.
We are sending the revised form of the manuscript entitled “Equine musculoskeletal pathologies: clinical approaches and therapeutical perspectives - a review”. The authors addressed, point by point, the reviewer’s questions. The changes are shown highlighted in yellow throughout the manuscript. We hope that this revised version of the manuscript fulfils all the requirements in order to be published in the Veterinary Sciences journal.
On behalf of all authors,
Ana Colette Maurício
Professora Catedrática/ Full Professor
Diretora do Departamento de Clínicas Veterinárias
Diretora do Doutoramento em Ciências Veterinárias
ICBAS - Universidade do Porto
Rua Jorge Viterbo Ferreira, nº 228
4050-313 Porto, Portugal
TEL: +351 220 428 000; TELM: +351 919071286
EXT: 5367 (Gab), 5371 (Lab)
ana.colette@hotmail.com; acmauricio@icbas.up.pt
Reviewer #3
- Figure 1 is too basic and unnecessary.
Thank you for your comment. In fact, it is. However, its incorporation was solely due to the attempt to quickly and concisely clarify which pathologies are addressed in the present work.
- All the information contained in the headings: 2) epidemiology; 3) most common musculoskeletal pathologies; 4) clinical examination; 5) complementary diagnostic exams - is not directed towards the objective of the review. At no point, in this vast expanse of text, do they address treatments. I therefore strongly urge the authors to consider deleting all the information contained here and, after a brief introduction, to start describing the treatment options.
Thank you for your comment.
It is a fact that the focus of this work is to address the therapies used in musculoskeletal injuries in horses. However, the title and introduction also refer to “clinical approaches”.
It seems pertinent to us to have a brief overview of epidemiology, clinical exam and precise diagnostic exams before introducing treatment options. We believe that this integrative approach makes sense in order to highlight some concepts in order to enable the understanding of therapeutic approaches.
Also, a review of the main diagnostic exams is vital as they are paramount to control treatments effectiveness.
Additionally, reviewer #2, kindly asked us to add a different and recent complementary exam that was not addressed in this work.
In fact, adopting this approach does result in a longer article, but given that it is a review, its extended length is inherent. We aimed to cover the topic comprehensively.
- Table 1: In my opinion, and generally in the medical literature, surgery is not considered a conservative treatment, but an interventional one. Consider dividing the table into 3 columns with conservative treatment, surgical treatment and regenerative treatment. In the table's legend, it is necessary to name the acronyms present, for example AI's; HyA; PRP; etc.
Thank you for your comment.
As a matter of fact, this inclusion might lead to that concern. The reason why “tendon splitting” was included as surgical treatment in the group of conservative treatments owes to, although surgical and interventional, the outcome is conservative since there is no tissue regeneration. However, it is now separated to avoid this kind of issue.
Table’s legend was also rewritten - lines 318-331.
- Lines 318-330: this information repeats literally what is present in
the table. Delete
Corrected. This information is now the table’s legend.
- Lines 346-352: Again, this information is redundant with the rest of the manuscript. If you're going to explore each of these topics ahead, there's no need for more text on them. The manuscript is too big to repeat information unnecessarily.
Thank you for your pertinent comment. This part was deleted.
- Lines 353-358: This information is already contained in lines 332-336, since physiotherapeutic modalities are part of Conservative therapies.
Deleted.
- Lines 375-377: Already mentioned below. Delete
Deleted.
- Line 378 and line 401: cold therapy and heat therapy are not numbered separately as a subtitle. So either the authors number them or they should appear in running text and not separated into sections.
Corrected. It was decided to also number these sections – lines 363 and 386.
- Line 407: "Heating prior to exercise...." this isn't therapy, it's injury prevention. This is done on any horse with or without pathology. Prevention and treatment are different matters. The aim here is to summarize the role of each therapy in the treatment of various pathologies and not their role in preventing disease. The authors have to decide whether they want to talk about the preventive role or the therapeutic role. Addressing both topics makes the text too long. This kind of mixing of concepts (treatment vs prevention) occurs throughout the text including discussion. Please revise the manuscript accordingly and eliminate these misconceptions.
Corrected.
- Lines 437-443: once again, the authors are talking about diagnosis which is not the focus of the work
The comment was addressed.
- Line 485: table 3 is not present
Thank you. This formatting lapse was addressed. Table 3 is present at line 465.
- Each treatment section has too much text/information. Giving the example of the laser, it is not the aim of this work to explain what the laser is or its principles. There is specialized literature on this. We want to know in which pathologies it is applicable, what effects it causes (and it is possible to summarize this part, since it is not necessary to explain the entire molecular perspective of its effects...if it reduces inflammation, it is not necessary to
address each component of the inflammatory cascade...clinically it is not interesting to go into so much detail) and how it should be used (protocols for example).
Thank you for your comment. All of this part was revised and shortened.
It would be very interesting to put more protocols but they vary a lot between different pathologies. Also, when we are talking about electrical devices, usually manufacturers include the protocols regarding each machine characteristics. However, it is generally described how each treatment should be used/done.
- Lines 813-834: it's a huge paragraph! It talks about diagnosis between lines 820-823 and doping in lines 825-834. There's no emphasis on the therapeutic component...
Thank you for your comment. The paragraph was rewritten.
A study conducted in standardbred race horses with fetlock traumatic osteoarticular lesions demonstrated that IV treatment of tiludronate in 500 mL of saline solution decreased inflammatory process and cartilage degeneration after treatment, meaning it inhibited the radiographic progression of OA in fetlocks at 6 months after treatment by inhibiting subchondral bone remodelling [155]. It was also highlighted the advantage of using tiludronate in young horses to control subchondral bone pain in the initial stages of OA, because it causes an acidic environment near the osteoclasts, inhibiting free nerve endings activation [155]. It is consensual, bisphosphonates have analgesic effects.
- Line 848: "6.1.3 Conservative surgical technique" - in my opinion "conservative surgical..." is a contradiction.
Corrected in the table and a sub-section was added - line 841.
- Once again, the lack of synthesis is obvious. The text between lines 1355-1373 has no relevance, it talks about prevention and even orthopaedic management of foals... This is a great mixture of information that confuses and makes the text difficult to read.
Before evaluating the rest of the discussion in detail, I would ask the authors to delete the paragraphs/sections mentioned by the reviewer and to review the entire manuscript in order to eliminate what is not relevant and, above all, to synthesize the information contained.
I understand that clinically there is an interest in approaching everything in an integrated way, but scientifically it is not correct when there is a pre-defined scientific objective.
Thank you very much for your comment. Some paragraphs were rewritten and others deleted in order to turn this part more factual.
- Please check the references (references number 25)
Thank you for your pertinent comment and the references were corrected and checked.

Round 2
Reviewer 3 Report
Comments and Suggestions for Authors
The reviewer congratulates the authors on the changes they have made, but the manuscript still has changes to be made.
I accept the fact that, with the title "clinical approaches and therapeutical perspectives", the sections on "clinical examination" and "complementary diagnostic exams" should be included. However, the sections on "epidemiology" and "Most common musculoskeletal pathologies" must be deleted. They take up a large part of the manuscript and add nothing. If you think about it, the clinical approach to a horse with lameness (except grade 5) should always be the same regardless of this epidemiological information. In the treatment sections, this information is not important either. If you talk about tendonitis and OA in these sections, you don't have to explain what they are or why they're important because most readers already know about them. You dont need either to do previous considerations about these diseases. As so, these two sections should be deleted
table 1: too much information in the legend. in the previous revision I asked for the information that was duplicated between text and table to be eliminated and not for this information to be included in the legend. Therefore, the table legend should only include its identification ("Therapeutical options for musculoskeletal pathologies") and the identification of acronyms (AI's, NSAID's....)
Laser section: yet too big and with particularities of no interest to this work.
Lines 700-728: the lack of ability to synthesize is still very visible. The characteristics and action of HyA are constantly repeated (lubrication, anti-inflammatory, effects on horses with OA....). Information such as "the first study carried out on seventies...." is information that is not relevant. As a rule, books have a historical context. A scientific review article should summarize the current state of the art in a brief way.
The same problem occur in the biphosphonates section (6.1.2.6)...
In my opinion, the sections of the treatments that could be written with details are regenerative therapies, because it's a "hot topic" of research but the details and how it works aren't so well known by the general clinical practitioners.
Author Response
Porto, 12 of April 2024
Dear Editors
Dear Reviewers
Veterinary Sciences
Title: Equine musculoskeletal pathologies: clinical approaches and therapeutical perspectives - a review. Authors: I.L. Reis, B. Lopes, P. Sousa, A.C. Sousa, A.R. Caseiro, C.M. Mendonça, J.M. Santos, L.M. Atayde, R.D. Alvites, A.C. Maurício.
We are sending the revised form of the manuscript entitled “Equine musculoskeletal pathologies: clinical approaches and therapeutical perspectives - a review”. The authors addressed, point by point, the reviewer’s questions. The changes are shown highlighted in yellow throughout the manuscript. We hope that this revised version of the manuscript fulfils all the requirements in order to be published in the Veterinary Sciences journal.
On behalf of all authors,
Ana Colette Maurício
Professora Catedrática/ Full Professor
Diretora do Departamento de Clínicas Veterinárias
Diretora do Doutoramento em Ciências Veterinárias
ICBAS - Universidade do Porto
Rua Jorge Viterbo Ferreira, nº 228
4050-313 Porto, Portugal
TEL: +351 220 428 000; TELM: +351 919071286
EXT: 5367 (Gab), 5371 (Lab)
ana.colette@hotmail.com; acmauricio@icbas.up.pt
Reviewer #3
- The reviewer congratulates the authors on the changes they have made, but the manuscript still has changes to be made. I accept the fact that, with the title "clinical approaches and therapeutical perspectives", the sections on "clinical examination"and "complementary diagnostic exams" should be included. However, the sections on "epidemiology" and "Most common musculoskeletal pathologies" must be deleted. They take up a large part of the manuscript and add nothing. If you think about it, the clinical approach to a horse with lameness (except grade 5) should always be the same regardless of this epidemiological information. In the treatment sections, this information is not important either. If you talk about tendonitis and OA in these sections, you don't have to explain what they are or why they're important because most readers already know about them. You don’t need either to do previous considerations about these diseases. As so, these two sections should be deleted.
It was deleted as suggested.
- table 1: too much information in the legend. in the previous revision I asked for the information that was duplicated between text and table to be eliminated and not for this information to be included in the legend. Therefore, the table legend should only include its identification ("Therapeutical options for musculoskeletal pathologies") and the identification of acronyms (AI's, NSAID's....).
Thank you for your comment. In fact, the information was deleted from the body of the text and only included in the legend. It is now reformulated.
- Laser section: yet too big and with particularities of no interest to this work.
This section was reformulated.
Lines 700-728: the lack of ability to synthesize is still very visible. The characteristics and action of HyA are constantly repeated (lubrication, anti-inflammatory, effects on horses with OA....). Information such as "the first study carried out on seventies...." is information that is not relevant. As a rule, books have a historical context. A scientific review article should summarize the current state of the art in a brief way. The same problem occur in the biphosphonates section (6.1.2.6)...
HyA and Biphosphonates were reviewed and summarized.
In my opinion, the sections of the treatments that could be written with details are regenerative therapies, because it's a "hot topic" of research but the details and how it works aren't so well known by the general clinical practitioners.
Thank you very much for your comment. I was followed your suggestion.

Round 3
Reviewer 3 Report
Comments and Suggestions for Authors
Thanks for the corrections
I have no more comments